
**Characterization of Aerosol Composition, Aerosol Acidity and Organic Acid Partitioning**
**at an Agriculture-Intensive Rural Southeastern U.S. Site**
Theodora Nah,[1] Hongyu Guo,[1] Amy P. Sullivan,[2] Yunle Chen,[1] David J. Tanner,[1] Athanasios
Nenes,[1,3,4,5] Armistead Russell,[6] Nga Lee Ng,[1,3] L. Gregory Huey[1] and Rodney J. Weber[1,*]
*[1]School of Earth and Atmospheric Sciences, Georgia Institute of Technology, Atlanta, GA, USA*
*[2]Department of Atmospheric Science, Colorado State University, Fort Collins, CO, USA*
*[3]School of Chemical and Biomolecular Engineering, Georgia Institute of Technology, Atlanta, GA, USA*
*[4]ICE-HT, Foundation for Research and Technology, Hellas, 26504 Patras, Greece*
*[5]IERSD, National Observatory of Athens, P. Penteli, 15236, Athens, Greece*
*[6]School of Civil and Environmental Engineering, Georgia Institute of Technology, Atlanta, GA, USA*
*\* To whom correspondence should be addressed: rweber@eas.gatech.edu*
**Abstract**
The implementation of stringent emission regulations has resulted in the decline of
anthropogenic pollutants including sulfur dioxide ($SO_2$), nitrogen oxides ($NO_x$) and carbon
monoxide (CO). In contrast, ammonia ($NH_3$) emissions are largely unregulated, with emissions
projected to increase in the future. We present real-time aerosol and gas measurements from a
field study conducted in an agricultural-intensive region in the southeastern U.S. during the fall
of 2016 to investigate how $NH_3$ affects particle acidity and SOA formation via the gas-particle
partitioning of semi-volatile organic acids. Particle water and pH were determined using the
ISORROPIA-II thermodynamic model and validated by comparing predicted inorganic $HNO_3$-
$NO_3^-$ and $NH_3$-$NH_4^+$ gas-particle partitioning ratios with measured values. Our results showed
that despite the high $NH_3$ concentrations (study average $8.1 \pm 5.2$ ppb), $PM_1$ were highly acidic
with pH values ranging from 0.9 to 3.8, and a study-averaged pH of $2.2 \pm 0.6$. $PM_1$ pH varied by
approximately 1.4 units diurnally. Formic and acetic acids were the most abundant gas-phase
organic acids, and oxalate was the most abundant particle-phase water-soluble organic acid
anion. Measured particle-phase water-soluble organic acids were on average 6 % of the total
non-refractory $PM_1$ organic aerosol mass. The measured molar fraction of oxalic acid in the
particle phase (i.e., particle-phase oxalic acid molar concentration divided by the total oxalic acid
molar concentration) ranged between 47 and 90 % for $PM_1$ pH 1.2 to 3.4. The measured oxalic
acid gas-particle partitioning ratios were in good agreement with their corresponding
thermodynamic predictions, calculated based on oxalic acid's physicochemical properties,
ambient temperature, particle water and pH. In contrast, gas-particle partitioning of formic and



acetic acids were not well predicted for reasons currently unknown. For this study, higher $NH_3$
concentrations relative to what has been measured in the region in previous studies had minor
effects on $PM_1$ organic acids and their influence on the overall organic aerosol and $PM_1$ mass
concentrations.
**1. Introduction**
Ammonia ($NH_3$) is the most abundant basic gas in the troposphere and plays an important
role in many atmospheric processes. It is a major neutralizer of atmospheric acidic species,
reacting readily with sulfuric acid ($H_2SO_4$) and nitric acid ($HNO_3$) to form ammonium sulfate
and nitrate salts (e.g., $(NH_4)_2SO_4$, and other forms such as $NH_4HSO_4$, $(NH_4)_3H(SO_4)_2$, and
$NH_4NO_3$), which are often the main inorganic components of atmospheric aerosols. Wet and dry
deposition are the principle $NH_3$ sinks (Dentener and Crutzen, 1994). $NH_3$ is spatially
heterogeneous, with the highest concentrations typically found near emission sources (Seinfeld
and Pandis, 2016). The dominant $NH_3$ sources in rural areas are agricultural in nature, and
include the application of fertilizers and volatilization of livestock waste (Reis et al., 2009; Ellis
et al., 2013; Van Damme et al., 2014). Biomass burning, either from wildfires or from controlled
burning during land-clearing operations, is also a significant source of $NH_3$ in rural
environments. The primary source of $NH_3$ in urban areas are industrial emissions, though
vehicular emissions can be a significant $NH_3$ source in some heavily populated cities (Reis et al.,
2009; Yao et al., 2013; Sun et al., 2017).
In the US, implementation of stringent emission controls on traditional anthropogenic air
pollutants, such as sulfur dioxide ($SO_2$), nitrogen oxides ($NO_x$) and carbon monoxide (CO), have
led to steady decreases in their emissions, and consequently their concentrations (Blanchard et
al., 2013a; Xing et al., 2013). In contrast, $NH_3$ emissions are largely unregulated, and are
projected to increase due to increased agricultural operations to feed a growing world population
(Reis et al., 2009; Ellis et al., 2013). Satellite observations showed that gas-phase $NH_3$
concentrations have increased substantially in US agricultural areas from 2002 to 2014 (Warner
et al., 2017). More wildfires from a changing climate, or from controlled burning for land
clearing for agricultural use, may also lead to increased $NH_3$ emissions (Reis et al., 2009;
Pechony and Shindell, 2010; Warner et al., 2016). These trends suggest that $NH_3$ could play an
increasingly important role in atmospheric chemistry.





Previous laboratory studies have shown that $NH_3$ can influence secondary organic aerosol
(SOA) formation and processing. For example, $NH_3$ increases SOA mass yields in the α-pinene
ozonolysis system, and is hypothesized to be due to the formation of ammonium salts from the
reaction of $NH_3$ with organic acids (Na et al., 2007). The heterogeneous uptake of $NH_3$ by SOA
can also lead to the formation of particulate organonitrogen compounds, a class of brown carbon
species that can reduce visibility and impact climate (Laskin et al., 2010; Updyke et al., 2012;
Lee et al., 2013; Laskin et al., 2015).
The southeastern U.S. is a natural outdoor laboratory for studying the effects of biogenic-
anthropogenic interactions on atmospheric aerosol formation and processing. Subtropical
vegetation composed mainly of mixed conifer and deciduous forests emit large quantities of
biogenic volatile organic compounds (BVOCs) that can act as precursors for SOA formation
(Blanchard et al., 2011; Guenther et al., 2012; Blanchard et al., 2013b). Large urban centers and
small towns are surrounded by large expanses of forests and widespread rural areas with
agricultural activities. Scattered within the southeastern U.S. are also coal-burning power plants
and industrial facilities. Anthropogenic activities in this region emit large concentrations of
VOCs, $SO_2$, $NO_x$, CO, $NH_3$ and aerosols (Blanchard et al., 2013c). Similar to other parts of the
U.S., $SO_2$, CO and $NO_x$ concentrations have decreased steadily in the southeastern U.S. due to
the implementation of emission controls (Blanchard et al., 2013a). In contrast, gas-phase $NH_3$
concentrations have increased in the southeastern U.S. over the same time period (Saylor et al.,
2015). These factors make the southeastern U.S. an intriguing place to study the influence of
$NH_3$ on atmospheric aerosol chemistry.
We performed aerosol and gas measurements during a field study conducted in Yorkville,
Georgia, U.S., in the fall of 2016, with the goal of understanding how $NH_3$ affects aerosol acidity
and SOA formation. The field site is surrounded by forest and agricultural land, affording an
opportunity to make ambient observations in an area impacted by local emissions of BVOCs and
$NH_3$. In this paper, we present gas and aerosol composition measurements that includes a suite of
organic acids. The thermodynamic equilibrium model, ISORROPIA-II, is used to calculate
particle water and pH based on measured inorganic aerosol and gas composition (Nenes et al.,
1998; Fountoukis and Nenes, 2007), and these predictions are compared to observed gas-particle
partitioning of $NH_3$, $HNO_3$ and organic acids. Together, these measurements are used to



determine how aerosol acidity affects the mass concentration of particle-phase organic acids at
this site.

## 2. Methods

### 2.1. Field site

Aerosol and gas measurements were conducted at the Yorkville, Georgia (33.929 N,
85.046 W) SouthEastern Aerosol Research and Characterization (SEARCH) field site from mid-
August to mid-October 2016. This is one of the sampling sites for the Southeastern Center for
Air Pollution and Epidemiology (SCAPE) study where aerosol characterization measurements
were conducted in the summer and winter of 2012 (Xu et al., 2015a; Xu et al., 2015b). A detailed
description of the field site can be found in Hansen et al. (2003). This rural site is situated in a
mixed forest-agriculture area approximately 55 km northwest and generally upwind of Atlanta.
The immediate surrounding area is used for cattle grazing and poultry concentrated animal
feeding operations (CAFOs) (Fig. S1). There are no major roads near the field site and nearby
traffic emissions were negligible. A large coal-fired power plant (Plant Bowen) is situated
approximately 25 km north of the site. Hence, the field site is impacted mainly by BVOC and
$NH_3$ emissions, with occasional spikes in $SO_2$ and minimal influence from urban anthropogenic
pollutants such as $HNO_3$, $O_3$, $NO_x$ and CO (Fig. S2). The sampling period was characterized by
moderate temperatures (24.0 °C average, 32.6 °C max, 9.5 °C min) and high relative humidities
(68.9 % RH average, 100 % RH max, 21.6 % RH min). Meteorological data are shown in Fig.
S3. Data reported are displayed in eastern daylight time (EDT).

### 2.2. Instrumentation

Instruments were housed in a temperature controlled (~20 °C) trailer during the field
study. Gas-phase $HNO_3$, $SO_2$ and organic acids (formic, acetic, oxalic, butyric, glycolic,
propionic, valeric, malonic and succinic acids) were measured by a custom-built chemical
ionization mass spectrometer (CIMS) using sulfur hexafluoride ions ($SF_6^-$) as reagent ions. $SO_2$
and $HNO_3$ were detected as fluoride adducts ($F_2SO_2^-$ and $NO_3^- \cdot HF$, respectively) while the
organic acids (HX) were detected primarily as conjugated anions ($X^-$) by the quadrupole mass
spectrometer (Huey et al., 1995; Huey et al., 2004; Nah et al., 2018). This CIMS is referred
hereafter as the $SF_6$-CIMS. Gas-phase $NH_3$ was measured by an additional custom-built CIMS





using protonated ethanol clusters $((C_2H_5OH)_n^+)$ as reagent ions. $NH_3$ was detected primarily as
$NH_4^+$ ions by the quadrupole mass spectrometer (Nowak et al., 2002; Yu and Lee, 2012; You et
al., 2014a). This CIMS is referred hereafter as the $NH_3$-CIMS.
Since $HNO_3$, $NH_3$ and organic acids may condense on surfaces, both $SF_6$-CIMS and
$NH_3$-CIMS used inlet configurations that minimized wall interactions (Huey et al., 2004; Nowak
et al., 2006). Each CIMS was connected to an inlet (a 7.6 cm ID aluminum pipe) that protruded
beyond the trailer's wall by ~40 cm into the ambient air. Both inlets were ~2 m above the
ground. A donut-shaped ring was attached to the ambient sampling port of each pipe to curtail
the influence of crosswinds on the pipe's flow dynamics. Both rings were wrapped with a fine
wire mesh to prevent ingestion of insects. A flow of ~2800 L min$^{-1}$ was maintained in each pipe
using regenerative blowers (AMETEK Windjammer 116637-03). Part of this flow (7 L min$^{-1}$ for
the $SF_6$-CIMS and 4.6 L min$^{-1}$ for the $NH_3$-CIMS) was sampled through a custom-made three-
way PFA Teflon valve, which connected the pipe's center to the CIMS sampling orifice and
could be switched automatically between ambient and background measurements.
Background measurements were performed every 25 min for 4 min for both the $SF_6$-
CIMS and $NH_3$-CIMS. During each background measurement, the sampled air flow was passed
through an activated charcoal scrubber (Sigma Aldrich) that removed $SO_2$, $HNO_3$ and organic
acids prior to delivery into the $SF_6$-CIMS, and through a silicon phosphate scrubber (Perma Pure
Inc.) that removed $NH_3$ prior to delivery into the $NH_3$-CIMS. > 99 % of the targeted species were
removed during background measurements for both the $SF_6$-CIMS and $NH_3$-CIMS. Standard
addition calibrations were performed every 5 h for the $SF_6$-CIMS using the outputs of a 1.12
ppm $^{34}SO_2$ gas cylinder (Scott-Marrin Inc.) and a formic or acetic acid permeation device (VICI
Metronics). Calibrations for the other gases measured by the $SF_6$-CIMS were performed in post-
field laboratory work, details of which can be found in Nah et al. (2018) and SI section S1.
Standard addition calibrations were performed hourly for the $NH_3$-CIMS using the output of a
$NH_3$ permeation device (KIN-TEK). The outputs of the formic and acetic acid permeation
devices were measured periodically by scrubbing the output of the permeation tube in deionized
water, followed by ion chromatography analysis for formate and acetate. The emission rate of the
$NH_3$ permeation device was measured using UV optical absorption (Neuman et al., 2003).





The detection limits for species measured by the $SF_6$-CIMS and $NH_3$-CIMS were
approximated from 3 times the standard deviation values ($3\sigma$) of the ion signals measured during
background mode. The detection limits for $HNO_3$, $SO_2$ and the various organic acids measured
by the $SF_6$-CIMS ranged from 1 to 60 ppt for 2.5 min integration periods, which corresponded to
the length of a background measurement with a 0.04 s duty cycle for each $m/z$ (Table S1).
Measurement uncertainties for the concentrations of $HNO_3$, $SO_2$ and the various organic acids
originate mainly from calibration measurements, and were between 12 and 25 % (Table S1). The
detection limit for $NH_3$ measured by the $NH_3$-CIMS was 1 ppb for 2.3 min integration periods,
which corresponded to the length of a background measurement with a 0.29 s duty cycle for the
$NH_4^+$ ion. Measurement uncertainties for $NH_3$ concentrations were 13 %.
A high-resolution time-of-flight aerosol mass spectrometer (HR-ToF-AMS, Aerodyne
Research Inc.) was used to measure the elemental composition of ambient non-refractory $PM_1$
(particles with aerodynamic diameters < 1 µm). Ambient air was sampled at 16.7 L min$^{-1}$ though
a URG $PM_1$ cyclone and then through a nafion dryer prior to delivery into the HR-ToF-AMS.
Aerosols were dried to RH < 20 % to eliminate the influence of RH on the HR-ToF-AMS's
particle collection efficiency. A detailed description of the HR-ToF-AMS can be found in the
literature (DeCarlo et al., 2006; Canagaratna et al., 2007; Canagaratna et al., 2015). Briefly, the
aerodynamic lens of the HR-ToF-AMS focused the dried submicron aerosols into a narrow
beam. The aerosols were then impacted onto a heated tungsten surface (~600 °C) where they
were flash vaporized. The resulting vapors were ionized by electron impact ionization (70 eV),
and the ions were detected by a time-of-flight mass spectrometer. Gas-phase interferences were
accounted for by subtracting the signals obtained during daily measurements of filtered, particle-
free sampling air. Ionization efficiency calibrations were performed weekly using 300 nm
ammonium nitrate and ammonium sulfate particles. Composition-dependent collection
efficiencies were applied to the data using the procedure detailed by Middlebrook et al. (2012).
Uncertainties in HR-ToF-AMS measurements were estimated to be approximately 25 %
(Canagaratna et al., 2007).
Particle-phase water-soluble organic acids, inorganic cations and anions were measured
using two Particle-into-Liquid Sampler (PILS) systems coupled to ion chromatographs (ICs)
(Orsini et al., 2003). Each PILS sampled ambient air at nominally 16.7 L min$^{-1}$ through a URG





185 $PM_1$ cyclone. Before PILS1, which was used to measure water-soluble inorganic cation and

186 anions, two long (24 cm) URG glass annular denuders coated with sodium carbonate and

187 phosphorous acid were used to remove acidic and basic gases. Before PILS2, which measured

188 water-soluble organic acids, a 28 cm parallel plate carbon denuder (Sunset Lab) was used to

189 remove organic gases (Eatough et al., 1993). In each PILS, aerosols were mixed with water

190 vapor at ~100 °C generated from heated ultrapure deionized water (Weber et al., 2001; Orsini et

191 al., 2003). The resulting droplets were impacted onto a plate, with the resulting liquid sample

192 analyzed by ICs. Each IC system was calibrated at the beginning and end of the study using five

193 multi-compound standards in order to create calibration curves. Periodically, a HEPA filter (Pall

194 Life Sciences) was placed on the inlet to determine the background in near real-time. The

195 measurement uncertainty for each IC system was about 10 %.

196  PILS1 was connected to two Dionex ICS-1500 ICs (Thermo Fisher Scientific) to measure

197 the water-soluble inorganic ions. These two IC systems include an isocratic pump, self-

198 regenerating anion or cation suppressor, and conductivity detector. This system will be referred

199 hereafter as the PILS-IC. Anions were separated using a Dionex IonPac AS15 guard and

200 analytical column (4 x 250 mm, Thermo Fisher Scientific) employing an eluent of 38 mM

201 sodium hydroxide at a flow rate of 1.5 mL $min^{-1}$. Cations were separated using a Dionex IonPac

202 CS12A guard and analytical column (4 x 250 mm, Thermo Fisher Scientific) employing an

203 eluent of 18 mM methanesulfonic acid at a flow rate of 1 mL $min^{-1}$. A new chromatogram was

204 obtained every 30 min with a sample loop fill time (i.e., ambient sample integration time) of 20

205 min. The limit of detection for the various anions and cations was approximately 0.01 μg $m^{-3}$.

206  PILS2 was coupled to a Dionex ICS-4000 capillary high-pressure ion chromatography

207 (HPIC) system to measure the water-soluble organic acids. The HPIC includes an eluent

208 generator, isocratic pump, degausser, suppressor, carbonate removal device, and conductivity

209 detector. This system will be referred hereafter as the PILS-HPIC. The organic acids were

210 separated using a Dionex AS11-HC-4μm capillary guard and analytical column (0.4 x 250mm,

211 Thermo Fisher Scientific), which used a potassium hydroxide gradient separation method at a

212 flow rate of 0.015 mL $min^{-1}$. A new chromatogram was obtained every 60 min with a sample

213 loop fill time of 2 min. The limit of detection for the various organic acids was approximately

214 0.001 μg $m^{-3}$.



Particle- and gas-phase water-soluble organic carbon ($WSOC_p$ and $WSOC_g$, respectively)
were measured using two Sievers 900 series total organic carbon (TOC) analyzers (GE
Analytical Instruments), as described by Sullivan et al. (2004). For $WSOC_p$ measurements,
ambient air was sampled at 15.2 L min$^{-1}$ through a URG $PM_1$ cyclone and a parallel plate carbon
denuder into a PILS coupled to the first TOC analyzer. For $WSOC_g$ measurements, ambient air
was sampled at 20 L min$^{-1}$ through a Teflon filter (45 mm diameter, 2.0 μm pore size, Pall Life
Sciences) to remove particles in the air stream. This filter was changed every 3 to 4 days. The
particle-free air was then directed to a MIST chamber filled with ultrapure deionized water,
which scrubbed the soluble gases at an air flow rate of 20 L min$^{-1}$. Soluble gases with Henry's
law constants greater than $10^3$ mole L$^{-1}$ atm$^{-1}$ were scrubbed into deionized water in the MIST
chamber (Spaulding et al., 2002). The resulting MIST chamber liquid sample was analyzed by
the second TOC analyzer. The TOC analyzers converted the organic carbon in the liquid samples
to carbon dioxide using UV radiation and chemical oxidation. The carbon dioxide formed was
then measured by conductivity. The amount of organic carbon in the liquid samples is
proportional to the measured increase in conductivity of the dissolved carbon dioxide. Each
$WSOC_p$ and $WSOC_g$ measurement lasted 4 min. Background $WSOC_p$ and $WSOC_g$
measurements were performed for 45 min every 12 h by stopping the sample air flow and rinsing
the system with deionized water. Both TOC analyzers were calibrated at the beginning and end
of the study using varying concentrations of sucrose solutions to create calibration curves (as
specified by the instrument manual). The limit of detections for $WSOC_p$ and $WSOC_g$ were 0.2
and 0.4 μgC m$^{-3}$, respectively. The measurement uncertainties for $WSOC_p$ and $WSOC_g$ were
estimated to be 10 % based on uncertainties in the TOC analyzer, sample air and liquid flows.
A suite of instruments operated by the SEARCH network provided supporting gas and
aerosol measurements (Hansen et al., 2003; Edgerton et al., 2005, 2006). $O_3$ was measured by a
UV absorption instrument (Thermo Fisher Scientific) with a temporal resolution of 1 min. NO
and $NO_x$ were measured by a chemiluminescence instrument (Thermo Fisher Scientific) with a
temporal resolution of 1 min. $NO_2$ was obtained from the difference between NO and $NO_x$. CO
was measured by a non-dispersive infrared absorption instrument (Thermo Fisher Scientific)
with a temporal resolution of 1 min. $NH_3$ was measured by a denuder-based instrument (ARA)
with a temporal resolution of 5 min. Comparisons of measurements by the $NH_3$-CIMS and
denuder-based instrument will be presented in section 3.1. A filter-based particle composition





monitor (ARA) provided 24 h-integrated PM$_{2.5}$ measurements of particle mass and major
inorganic ions measured offline by ion chromatography. Organic carbon (OC) and elemental
carbon (EC) in PM$_{2.5}$ were measured by a OCEC Analyzer (Sunset Labs) with a temporal
resolution of 1 h. This analyzer determined OC by thermal optical transmittance. VOCs were
measured by a gas chromatography-flame ionization detector (GC-FID, Agilent Technologies)
with a temporal resolution of 1h.

### 2.2. Particle pH and water calculation

The thermodynamic equilibrium model ISORROPIA-II was used to determine the phase

state and composition of an NH$_4^+$-SO$_4^{2-}$-NO$_3^-$-Cl$^-$-Na$^+$-Ca$^{2+}$-K$^+$-Mg$^{2+}$-water inorganic aerosol in
equilibrium with its corresponding gas-phase species (Fountoukis and Nenes, 2007; Nenes et al.,
1998). This approach was used in previous studies to determine particle water and pH in
different parts of the world (Guo et al., 2015; Bougiatioti et al., 2016; Guo et al., 2016; Weber et
al., 2016; Guo et al., 2017a; Guo et al., 2017c; Shi et al., 2017). pH is defined as the negative
logarithm of the hydronium ion (H$_3$O$^+$) activity in an aqueous solution. For simplicity, H$_3$O$^+$ is
denoted here as H$^+$ even though we recognize that the unhydrated hydrogen ion is rare in
aqueous solutions. The particle pH is calculated as:

$$pH = -\log_{10} \gamma_{H^+} H_{aq}^+ = -\log_{10} \frac{1000 \gamma_{H^+} H_{air}^+}{W_i + W_o} \cong -\log_{10} \frac{1000 \gamma_{H^+} H_{air}^+}{W_i} \qquad (1)$$

where $\gamma_{H^+}$ is the hydronium ion activity coefficient (assumed to be 1), $H_{aq}^+$ (mole L$^{-1}$) is the
molar concentration of hydronium ions in particle water (i.e., pH is defined in terms of molarity),
$H_{air}^+$ (µg m$^{-3}$) is the hydronium ion concentration per volume of air, and $W_i$ and $W_o$ (µg m$^{-3}$) are
the bulk particle water concentrations associated with inorganic and organic species,
respectively. In equation 1, the molecular weight of H$^+$ is taken as 1 g mole$^{-1}$, and 1000 is the
factor needed for unit conversion of g L$^{-1}$ to µg m$^{-3}$. $H_{air}^+$ and $W_i$ are outputs of the ISORROPIA-
II model. Previous studies have shown that particle pH values predicted using only $W_i$ are
reasonably accurate since the sensitivity of particle pH to the effects of $W_o$ is small (Guo et al.,
2015). For the southeastern U.S., Guo et al. (2015) reported that particle pH values predicted
using only $W_i$ were systematically 0.15 to 0.23 units lower than those predicted using $W_i + W_o$
during the 2013 Southern Oxidant Aerosol Study (SOAS) and SCAPE campaigns. Given this


small deviation and that organic aerosol hygroscopicity was not measured in this field study, we
report particle pH only considering $W_i$.

ISORROPIA-II was run in "forward" mode, which assumes that aerosols are

"metastable" with no solid precipitates, to predict particle pH and the partitioning of semi-
volatile compounds. In "forward" mode, the model calculates the gas-particle equilibrium
partitioning concentrations based on the input of the total concentration of a species (i.e., gas +
particle). In "reverse" mode, the model calculates the gas-particle equilibrium partitioning
concentrations based on the input of only the particle-phase concentration of a species. We used
"forward" mode because the "reverse" mode is sensitive to measurement errors, which often
result in large model biases in the predicted particle pH (Hennigan et al., 2015). The measured
particle-phase inorganic $NH_4^+$, $SO_4^{2-}$ and $NO_3^-$ concentrations and gas-phase $HNO_3$ and $NH_3$
concentrations were used as model inputs. The "metastable" assumption is reasonable since the
high RH (average RH 68.9 %) observed during the study indicated that the aerosols had likely
deliquesced. We excluded data for periods where the RH was above 95 % since the exponential
growth in particle liquid water with RH introduces large pH uncertainties (Malm and Day, 2001;
Guo et al., 2015).

In using ISORROPIA-II to predict particle pH and the partitioning of semi-volatile

compounds, we also assumed that the aerosols are internally mixed and that the particle pH does
not change with particle size (i.e., the overall particle pH is characterized by the particle's bulk
properties). As long as some small fraction of sulfate is mixed with various aerosol components,
(e.g., non-volatile cations), the assumption that aerosols are completely internally mixed has a
small effect on the predicted pH (Guo et al., 2017b). However, the presence of multiple organic
and inorganic species in ambient aerosols may lead to multiple phases within the particle (i.e.,
phase separation). Consequently, this may result in the unequal distribution of inorganic species
among different phases, each with its own water activity and inorganic concentration. Previous
studies have shown that liquid-liquid and solid-liquid phase separations may occur for mixed
organic and inorganic aerosols at low RH and organic aerosol oxygen-to-carbon atomic ratios
(O/C) (Bertram et al., 2011; Song et al., 2012; You et al., 2013; You et al., 2014b; You and
Bertram, 2015). Phase separations were always observed at O/C ≤ 0.5, while no phase separation
was observed at O/C ≥ 0.8. The probability for the occurrence of phase separation decreased at



higher RH for 0.5 < O/C < 0.8. The average O/C for this field study is 0.69 ± 0.06. Organic acids
were not included in the calculation of particle pH. This is reasonable since their total mass
concentration was small compared to the total inorganic mass concentration. The study-averaged
ratio of the organic acid mass concentration to the inorganic mass concentration is 0.25.
Furthermore, Song et al. (2018) showed that including organic acid mass concentrations in
thermodynamic model calculations had minor effects on particle pH if the system is in
equilibrium. The validity of these assumptions and the resulting thermodynamic model
predictions will be evaluated by comparing the predicted gas-particle partitioning ratios of semi-
volatile inorganic compounds with measured values in section 3.3.
**3. Results and Discussion**
**3.1. NH$_3$ observations**
Continuous measurements of NH$_3$ were made using the NH$_3$-CIMS from 13 September to
12 October. Figures 1a and 1b show the time series and study-averaged diurnal profile of NH$_3$,
respectively. NH$_3$ concentrations ranged from 0.7 to 39.0 ppb (0.5 to 28.5 μg m$^{-3}$), and exhibited
consistent diurnal cycles. NH$_3$ was generally higher in the late mornings and early afternoons.
Concentrations started to increase at 07:30, which coincided with an increase in temperature at
sunrise (Fig. S3). Possible reasons for the morning increase include volatilization of particulate
ammonium and animal waste, entrainment from the residual layer where NH$_3$ may not have been
depleted, evaporation of dew or fog that contained dissolved NH$_3$, and emission from plant
stomata (Ellis et al., 2011). NH$_3$ decreased at 14:30, approximately 1 hour before temperature
decreased, and may be due to changes in the boundary layer height. The diurnal plot does not
account for dilution as the boundary layer expanded, and only indicates that if emissions were
solely from the surface and lower concentrations aloft, these NH$_3$ sources were of significant
magnitude.
The average NH$_3$ concentration measured by the NH$_3$-CIMS is 8.1 ± 5.2 ppb. This is
approximately 2 times higher than the average NH$_3$ concentration (3.8 ± 2.9 ppb) measured by
the denuder-based instrument operated by the SEARCH network over the same time period (Fig.
S4). Differences in NH$_3$ concentrations measured by the two instruments may be due to positive
and negative sampling artifacts caused by differences in sampling inlets (e.g., inlet length and





location), frequency of calibration and background measurements, and (in the case of the denuder-based instrument) possible sample contamination during chemical analysis. Discussions on how differences in measured $NH_3$ concentrations affect $PM_1$ pH predictions will be presented in section 3.3. Nevertheless, there is a record of $NH_3$ concentrations measured by the denuder-based instrument at this site since 2008. Just prior to, and during this study, $NH_3$ concentrations are generally the highest observed since 2011 (Fig. S5). These elevated $NH_3$ concentrations may be due to sporadic biomass burning episodes caused by elevated temperatures and widespread drought across the southeastern U.S. in 2016 (Park Williams et al., 2017; Case and Zavodsky, 2018).

The $NH_3$-CIMS measurements are examined with the meteorological data to gain insights on the primary $NH_3$ sources during the sampling period. To account for wind speed, the 1-hour averaged $NH_3$ concentrations are first multiplied by their corresponding 1-hour averaged wind speeds. These normalized $NH_3$ concentrations are then used to construct a wind direction polar plot showing the average normalized $NH_3$ concentration per 10 degrees bin (Fig. 1c). The wind direction polar plot shows that the normalized $NH_3$ is approximately 2 times greater than the average when air masses are transported from the south-east, the general direction of the poultry CAFOs located approximately 2 km from the field site (Fig. S1), which are known for high $NH_3$ emissions. This conclusion is reaffirmed by $NH_3$ measurements by the SEARCH network's denuder-based instrument.

$NH_3$ concentrations measured by the two instruments in this study are substantially higher than those measured in three recent field studies conducted in the continental U.S.: 2010 California Nexus (CalNex) study, 2013 Southeast Nexus (SENEX) study and 2013 SOAS study (see Table 1). The differences in $NH_3$ may be attributed to differences in land use, proximity to CAFOs and meteorological conditions. The high $NH_3$ concentrations in this study allow us to make ambient observations of the effect of $NH_3$ on particle acidity and the gas-particle partitioning of semi-volatile inorganic and organic compounds, and compare them with previous studies.

**3.2. $PM_1$ composition**

The aerosol inorganic chemical composition was measured by several instruments during





this study. The HR-ToF-AMS, PILS-IC and PILS-HPIC measured the composition of $PM_1$,
while a filter-based particle composition monitor measured the composition of $PM_{2.5}$.
Comparisons of aerosol $SO_4^{2-}$, $NO_3^-$ and $NH_4^+$ mass concentrations measured by these four
instruments are summarized in Fig. S6. $NH_4^+$ measurements by the PILS-IC are not available for
comparison due to denuder breakthrough that occurred during the study.
$SO_4^{2-}$ measurements by the various instruments are generally well correlated with each
other, with $R^2$ values ranging from 0.64 to 0.92. Although $PM_1$ $SO_4^{2-}$ measurements by the two
PILS systems show good agreement with each other, HR-ToF-AMS $SO_4^{2-}$ measurements are
approximately two times higher than the PILS and filter measurements. Similar systematic
differences are also observed for $NO_3^-$ and $NH_4^+$ measurements. $NO_3^-$ and $NH_4^+$ measurements
by the four instruments are moderately correlated ($R^2 = 0.54$ to 0.79 and $R^2 = 0.94$, respectively).
$NO_3^-$ measurements by the PILS and filter systems are mostly similar; however, HR-ToF-AMS
$PM_1$ $NO_3^-$ and $NH_4^+$ measurements are approximately three times and two times higher than the
PILS and filter measurements. Although the higher HR-ToF-AMS $PM_1$ $NO_3^-$ measurements may
be due, in part, to the HR-ToF-AMS not being able to readily discriminate between the inorganic
and organic nitrates, reasons for the HR-ToF-AMS $PM_1$ $SO_4^{2-}$ and $NH_4^+$ measurements being
systematically higher than the PILS and filter measurements are not known.
We estimated HR-ToF-AMS $PM_1$ mass concentrations that would be consistent with
PILS and filter measurements by multiplying all the raw HR-ToF-AMS data by a constant factor
of 0.5 (i.e., average of the PILS-HPIC/HR-ToF-AMS and PILS-IC/HR-ToF-AMS $SO_4^{2-}$ slopes).
The scaled HR-ToF-AMS data is used in all our subsequent analyses.
Figure 2 shows the time series and study-averaged diurnal profiles of non-refractory $PM_1$
species. The study-averaged non-refractory $PM_1$ organics, $SO_4^{2-}$, $NO_3^-$ and $NH_4^+$ mass
concentrations are $5.0 \pm 2.3$, $1.6 \pm 0.4$, $0.2 \pm 0.1$ and $0.4 \pm 0.2$ µg m$^{-3}$, respectively. Organics are
the dominant non-refractory $PM_1$ species, accounting for $74.2 \pm 7.9$ % of the non-refractory $PM_1$
mass concentration during the field study. Organic aerosol mass concentration was slightly
higher at night, which is likely caused by changes in the boundary layer height, emission sources
and SOA formation processes (Xu et al., 2015b). Apportionment of organic aerosol sources will
be discussed in an upcoming publication. $SO_4^{2-}$ is the second most abundant non-refractory $PM_1$
species ($16.3 \pm 5.7$ % mass fraction), followed by $NH_4^+$ ($5.9 \pm 2$ % mass fraction) and $NO_3^-$ (3.6





± 2.2 % mass fraction). $SO_4^{2-}$ mass concentration peaked in the afternoon due to enhanced $SO_2$
photooxidation (Weber et al., 2003). The $NO_3^-$ mass concentration measured by the HR-ToF-
AMS is the nitrate functional group ($-ONO_2$) present on organic and inorganic nitrates. Hence,
the diurnal profile of the $NO_3^-$ mass concentration in Fig. 2 has contributions from both organic
and inorganic nitrates. The $NO_3^-$ mass concentration increased after sunset and peaked at sunrise
due to the formation of organic nitrates from nighttime $NO_3$ chemistry and increased gas-to-
particle partitioning of organic and inorganic nitrates as temperature decreased (Xu et al., 2015a;
Xu et al., 2015b). Quantification of organic nitrates based on HR-ToF-AMS and PILS-IC $PM_1$
$NO_3^-$ measurements will be discussed in a future publication. $NH_4^+$ mass concentration has
moderate diurnal variations with marginally higher concentrations in the afternoon, likely due to
the contrasting day/night phases of ammonium sulfate and ammonium nitrate formation. $SO_4^{2-}$,
$NO_3^-$ and $NH_4^+$ molar concentrations indicated that $NH_4^+$ is mainly associated with $SO_4^{2-}$ in $PM_1$.
**3.3. $PM_1$ pH predictions**
CIMS $HNO_3$ and $NH_3$ data, scaled HR-ToF-AMS $PM_1$ $SO_4^{2-}$ and $NH_4^+$ data, PILS-IC
$PM_1$ $NO_3^-$ and non-volatile cation ($Cl^-$, $Na^+$, $Ca^{2+}$, $K^+$ and $Mg^{2+}$) data, temperature and RH are
used as ISORROPIA-II model inputs to predict $PM_1$ $W_i$ and pH from 13 September to 6 October.
Figure 3 shows the time series and study-averaged diurnal profiles of ISORROPIA-predicted
$PM_1$ $W_i$ and pH. $PM_1$ are highly acidic with pH values ranging from 0.9 to 3.8, and a study-
averaged pH of 2.2 ± 0.6. The average $PM_1$ pH was 2.5 ± 0.6 during periods where the $NH_3$
concentration was higher than 13.3 ppb (i.e., study-averaged $NH_3$ concentration + 1 standard
deviation = 8.1 + 5.2 = 13.3 ppb). The $PM_1$ pH values in this study are generally similar to those
reported by Guo et al. (2015) at the same field site during winter 2012. Our observation that $PM_1$
are acidic despite the high $NH_3$ concentrations in this study is consistent with previous studies
showing that particle pH has weak sensitivities to wide $NH_3$ and $SO_4^{2-}$ mass concentration ranges
due to pH buffering caused by the partitioning of $NH_3$ between the gas and particle phases
(Weber et al., 2016; Guo et al., 2017c). This weak particle pH sensitivity also explains the small
changes in $PM_1$ pH values (about 10 % lower, Fig. S7) when $NH_3$ measurements by the
SEARCH network denuder-based instrument are used in ISORROPIA-II calculations (instead of
$NH_3$-CIMS measurements).



PM$_1$ pH varied by approximately 1.4 units throughout the day. $W_i$ has a study-averaged
value of 1.6 ± 1.7 µg m$^{-3}$. PM$_1$ $W_i$ and pH showed similar diurnal profiles, with both peaking in
the mid-morning and reaching their minima in the mid-afternoon. These diurnal trends are
consistent with those previously reported by Guo et al. (2015) for PM$_1$ measured during the
summer and winter in different parts of the southeastern U.S., and reaffirm that diurnal variation
in particle pH is driven by $W_i$ and not aerosol chemistry.
The average PM$_1$ pH for this study is about 1 unit higher than those for the SENEX and
SOAS campaigns (Table 1), and is likely due to the much higher abundance of NH$_3$ in this study.
The average NH$_3$ mass concentration in this study is approximately 49 times and 15 times higher
than those in the SENEX and SOAS campaigns, respectively. The average PM$_1$ pH for this study
is similar to that for the CalNex campaign even though the average NH$_3$ mass concentration in
this study is only approximately 4 times higher than that in the CalNex campaign (Guo et al.,
2017a). This can be explained by PM$_1$ SO$_4^{2-}$ and NO$_3^-$ mass concentrations at CalNex being
approximately 2 times and 18 times larger than those of this study, respectively. Aerosol
inorganic SO$_4^{2-}$ and NO$_3^-$ species are highly hygroscopic. The much higher NO$_3^-$ mass
concentrations in the CalNex campaign (due, in part, to high NO$_x$ emissions) increased particle
$W_i$ substantially, which diluted H$^+$ and raised particle pH, resulting in more gas-to-particle
partitioning of NO$_3^-$, and eventually leading to pH levels similar to those observed in this study.
This type of feedback does not happen in the southeastern U.S. where non-volatile SO$_4^{2-}$
dominates the uptake of particle water.
The validity of this study's thermodynamic model predictions is evaluated by comparing
the predicted gas-particle partitioning ratios of semi-volatile inorganic compounds (i.e., NO$_3^-$ and
NH$_4^+$) with measured values (Fig. S8). CIMS HNO$_3$ and NH$_3$ data, PILS-IC NO$_3^-$ and scaled
HR-ToF-AMS NH$_4^+$ data are used in this comparison. ε(NO$_3^-$) and ε(NH$_4^+$) are defined as the
particle-phase molar concentration divided by the total molar concentration (gas + particle), i.e.,
ε(NO$_3^-$) = NO$_3^-$ / (HNO$_3$ + NO$_3^-$) and ε(NH$_4^+$) = NH$_4^+$ / (NH$_3$ + NH$_4^+$). Predicted NH$_3$, NH$_4^+$ and
ε(NH$_4^+$) values are generally within 10 % of and are highly correlated (R$^2$ = 0.96 to 0.99) with
measured values (Fig. S8). While predicted HNO$_3$ values generally agreed with measurements,
substantial scatter can be seen between the predicted and measured values for NO$_3^-$ and ε(NO$_3^-$).
This scatter can be attributed, at least in part, to uncertainties brought about by the low PM$_1$ NO$_3^-$





mass concentrations and effects of coarse mode cations (e.g., $Na^+$, $Ca^{2+}$, $K^+$ and $Mg^{2+}$) on fine
mode $HNO_3$-$NO_3^-$ gas-particle equilibrium (i.e., $HNO_3$ can partition to both fine and coarse
modes, thereby affecting fine mode $NO_3^-$ concentrations; no such effect occurs for $NH_3$-$NH_4^+$
gas-particle equilibrium). In general, the overall good agreement between model predictions and
measurements indicated that our assumptions that aerosols are metastable (i.e., aerosols are
supersaturated aqueous droplets) with no phase separation for the thermodynamic calculations
are reasonable for the conditions of this study, and do not affect model predictions.

The molar fractions of $NO_3^-$ and $NH_4^+$ in the particle phase (i.e., $\varepsilon(NO_3^-)$ and $\varepsilon(NH_4^+)$)

measured in this study are compared with those measured during the CalNex, SENEX and SOAS
campaigns. Figure 4 shows the measured $\varepsilon(NO_3^-)$ and $\varepsilon(NH_4^+)$ values as a function of their
ISORROPIA-predicted particle pH for the various field studies. For each field study, only a
subset of the data is chosen for this comparison ($1 \leq W_i \leq 4$ µg m$^{-3}$ and 15 °C $\leq$ temperature $\leq 25$
°C) to reduce the effects of variability of $W_i$ and temperature on gas-particle partitioning for
comparison with the calculated S (or sigmoidal) curves, which are calculated based on $W_i = 2.5$
µg m$^{-3}$ and temperature = 20 °C. The S curves for $HNO_3$-$NO_3^-$ and $NH_3$-$NH_4^+$ partitioning as a
function of particle pH are also plotted as solid lines. The S curves are calculated based on the
solubility and dissociation of $NO_3^-$ and $NH_4^+$ species in water:

$$\varepsilon(NO_3^-) = \frac{H_{HNO_3}^* RTW_i \times 0.987 \times 10^{-14}}{\gamma_{H^+}\gamma_{NO_3^-}10^{-pH} + H_{HNO_3}^* RTW_i \times 0.987 \times 10^{-14}} \qquad (2)$$

$$\varepsilon(NH_4^+) = \frac{\frac{\gamma_{H^+}10^{-pH}}{\gamma_{NH_4^+}} H_{NH_3}^* RTW_i \times 0.987 \times 10^{-14}}{1 + \frac{\gamma_{H^+}10^{-pH}}{\gamma_{NH_4^+}} H_{NH_3}^* RTW_i \times 0.987 \times 10^{-14}} \qquad (3)$$

where $H_{HNO_3}^*$ and $H_{NH_3}^*$ (mole$^2$ kg$^{-2}$ atm$^{-1}$) are equilibrium constants and are the products of the
Henry's law constant and the dissociation constant of $HNO_3$ and $NH_3$, respectively, $R$ is the gas
constant (8.314 m$^3$ Pa K$^{-1}$ mol$^{-1}$), $T$ is temperature (K), and $\gamma_i$'s are activity coefficients. $H_{HNO_3}^*$
and $H_{NH_3}^*$ values at 20 °C are calculated using equations found in Clegg and Brimblecombe
(1990) and Clegg et al. (1998), respectively. Activity coefficients predicted by ISORROPIA-II
are $\gamma_{H^+-NO_3^-} = \sqrt{\gamma_{H^+}\gamma_{NO_3^-}} = 0.28$, $\gamma_{H^+} = 1$ and $\gamma_{NH_4^+} = 1$. Derivations of the analytically
calculated S curves for $\varepsilon(NO_3^-)$ and $\varepsilon(NH_4^+)$ in equations 2 and 3 can be found in Guo et al.





(2017a). As shown in Fig. 4, the measured $\varepsilon(NO_3^-)$ and $\varepsilon(NH_4^+)$ values for the four field studies
all generally converged on the calculated S curves. The higher particle pH values in this study
and the CalNex campaign relative to those for the SENEX and SOAS campaigns resulted in less
$NH_3$ and more $HNO_3$ partitioned to the particle phase, as predicted by these simple analytical
expressions. A similar analysis will be performed for the organic acids in section 3.5.
**3.4. WSOC and water-soluble organic acids**
The time series and study-averaged diurnal profiles of $WSOC_g$ and $WSOC_p$ are shown in
Fig. S9. The study-averaged $WSOC_g$ mass concentration ($3.6 \pm 2.7$ µgC m$^{-3}$) is roughly four
times higher than that of $WSOC_p$ ($1.0 \pm 0.6$ µgC m$^{-3}$). The diurnal profile of $WSOC_p$ is
somewhat flat, likely due to various organic aerosol sources having different water solubility and
diurnal cycles, and compensating each other throughout the day (Xu et al., 2015b; Xu et al.,
2017). In contrast, $WSOC_g$ displayed strong diurnal variations. $WSOC_g$ increased at 07:30,
which coincided with the sharp increase in solar irradiance (Fig. S3). $WSOC_g$ decreased at 21:30,
approximately 2 hours after sunset. Also shown in Fig. S9 are the time series and study-averaged
diurnal profile of the mass fraction of total WSOC in the particle phase, i.e., $F_p$ = $WSOC_p$ /
($WSOC_p$ + $WSOC_g$). The peak $F_p$ coincided with the minima of $WSOC_g$ at 07:30.
The study-averaged $WSOC_g$ and $WSOC_p$ ($3.6 \pm 2.7$ µgC m$^{-3}$ and $1.0 \pm 0.6$ µgC m$^{-3}$) are
slightly lower than those measured during the SOAS campaign (SOAS $WSOC_g$ = 4.9 µgC m$^{-3}$
and $WSOC_p$ = 1.7 µgC m$^{-3}$) (Xu et al., 2017). While the diurnal profiles of $WSOC_p$ in both
studies are flat, the diurnal profiles of $WSOC_g$ measured in the two studies are different. $WSOC_g$
measured in the SOAS study decreased at sunset, while $WSOC_g$ measured in this study
decreased 2 hours after sunset. Differences in $WSOC_g$ diurnal profiles in the two studies are
likely due to differences in emission sources as a result of different sampling periods (SOAS was
in early summer and this study was in early fall), land use and/or land cover. The ratio of
$WSOC_p$ to OC for this study was estimated at 30 %, but this comparison is imprecise because
$WSOC_p$ was $PM_1$ and OC was $PM_{2.5}$ (refer to Fig. S10 and SI section S2).
Figure 5 shows the time series of particle- and gas-phase concentrations of formic, acetic,
oxalic, malonic, succinic, glutaric and maleic acids. Their diurnal profiles are shown in Fig. 6.
Gas-phase measurements of glutaric and maleic acids are not available. Gas-phase measurements





of butyric, glycolic, propionic and valeric acids were also measured during the study and have
been presented in Nah et al. (2018), but will not be discussed here since their particle-phase
measurements are not available.
Assuming that all the measured organic acids are completely water-soluble, 30 % of the
$WSOC_g$ is comprised of these organic acids (Nah et al., 2018). Formic and acetic acids are the
most abundant gas-phase organic acids, with study averages of $2.2 \pm 1.6$ and $1.9 \pm 1.3$ µg m$^{-3}$,
respectively. The study-averaged carbon mass fraction of $WSOC_g$ comprised of formic and
acetic acids are 7 and 13 %, respectively. All the gas-phase organic acids displayed strong and
consistent diurnal cycles, with higher concentrations being measured during warm and sunny
days. Their concentrations start to increase at sunrise (at 07:30), building to a peak between
15:30 and 19:30, then decrease overnight.
Nah et al. (2018) previously showed that the measured gas-phase organic acids during the
study, including oxalic acid, likely have the same or similar sources. Poor correlations between
gas-phase organic acid concentrations and those of anthropogenic pollutants ($HNO_3$, $SO_2$, CO
and $O_3$) indicated that these organic acids are not due to anthropogenic emissions, and are likely
biogenic in nature. Biogenic emissions of gas-phase organic acids and/or their BVOC precursors
are elevated at high temperatures, resulting in higher organic acid concentrations during warm
and sunny days. Some of these gas-phase organic acids may also be formed in the particle phase
during organic aerosol photochemical aging, with subsequent volatilization into the gas phase.
The measured particle-phase water-soluble organic acids contributed on average 6 % to
the scaled HR-ToF-AMS-measured organic aerosol mass concentration. The study-averaged
carbon mass fraction of $WSOC_p$ comprised of these organic acids is 4 %. Previous studies have
shown that particle-phase organic acids found in rural environments are oxidation products of
gas-phase aliphatic monocarboxylic acids, which are formed in the photochemical oxidation of
biogenic unsaturated fatty acids and other BVOC precursors (Kawamura and Gagosian, 1987;
Kawamura and Ikushima, 1993; Kerminen et al., 2000; Kawamura and Bikkina, 2016). These
particle-phase organic acids can also be produced during the multiphase photochemical aging of
ambient organic aerosols (Ervens et al., 2004; Lim et al., 2005; Sorooshian et al., 2007;
Sorooshian et al., 2010).




Oxalate is the most abundant measured particle-phase water-soluble organic acid anion
(contributing a study-averaged 26 % to the total particle-phase organic acid mass concentration),
with mass concentrations ranging from 0.01 to 0.34 μg m$^{-3}$ and a study average of 0.07 ± 0.05 μg
m$^{-3}$. Acetate (study average of 0.06 ± 0.03 μg m$^{-3}$) and formate (study average of 0.05 ± 0.03 μg
m$^{-3}$) are the second and third most abundant measured particle-phase water-soluble organic acid
anions, respectively. Particle-phase formate, acetate and maleate showed weak diurnal variations,
and may be due, in part, to various emission sources having different diurnal cycles and
compensating each other throughout the day. Particle-phase oxalate, malonate and succinate
peaked in the mid- to late afternoon, while glutarate generally peaked in the mid-morning. This
suggests that while the production of these organic acids is photochemically-driven, they likely
have different BVOC precursors and/or different photochemical production pathways.
**3.5. Gas-particle partitioning of organic acids**
The online and simultaneous measurements of gas- and particle-phase organic acid mass
concentrations provided the opportunity to study gas-particle partitioning behavior of semi-
volatile organic compounds with respect to particle pH, as is more commonly done with semi-
volatile inorganic species (see section 3.3). Since formic, acetic and oxalic acids are the three
most abundant measured organic acids present in the gas and particle phases, we focus on the
gas-particle partitioning behaviors of these three organic acids. The study-averaged molar
fractions (± 1 standard deviation) of formic, acetic and oxalic acid in the particle phase (i.e.,
$\varepsilon(HCOO^-)$, $\varepsilon(CH_3CO_2^-)$ and $\varepsilon(C_2O_4^{2-})$) are 3.6 ± 3.6 %, 5.8 ± 5.0 % and 73.7 ± 9.8 %,
respectively. The uncertainties of these ratios for formic, acetic and oxalic acids are 16, 16 and
17 %, respectively, which are obtained from the propagation of their SF$_6$-CIMS and PILS-HPIC
measurement uncertainties.
**3.5.1. Oxalic acid**
To investigate the factors affecting oxalic acid gas-particle partitioning, the equation for
the S curve describing the dependence of oxalic acid gas-particle partitioning (i.e., $\varepsilon(C_2O_4^{2-})$ =
$C_2O_4^{2-}$ / ($C_2H_2O_4 + C_2O_4^{2-}$)) on particle pH is derived. As shown in SI section S3, the analytically
calculated S curve for $\varepsilon(C_2O_4^{2-})$ can be simplified to:



$$\varepsilon(C_2O_4^{2-}) \cong \frac{H_{C_2H_2O_4}W_iRT\left(\frac{\gamma_H+\gamma_{C_2HO_4^-}}{\gamma_{C_2H_2O_4}}10^{-pH}+K_{a1}\right)\times0.987\times10^{-14}}{\gamma_H+\gamma_{C_2HO_4^-}10^{-pH}+H_{C_2H_2O_4}W_iRT\left(\frac{\gamma_H+\gamma_{C_2HO_4^-}}{\gamma_{C_2H_2O_4}}10^{-pH}+K_{a1}\right)\times0.987\times10^{-14}}$$ (4)

where $H_{C_2H_2O_4}$ (mole $L^{-1}$ atm$^{-1}$) is the Henry's law constant for oxalic acid, $K_{a1}$ (mole $L^{-1}$) is the
first acid dissociation constant for oxalic acid, $R$ is the gas constant (8.314 m$^3$ Pa K$^{-1}$ mol$^{-1}$), $T$ is
temperature (K), and $\gamma_i$'s are activity coefficients. We used the web version of AIOMFAC
(www.aiomfac.caltech.edu) (Zuend et al., 2008; Zuend et al., 2011; Zuend et al., 2012) to
compute a study-averaged $\gamma_{C_2H_2O_4}$ value of 0.0492. Since AIOMFAC does not predict
$\gamma_H+\gamma_{C_2HO_4^-}$, we assumed that $\gamma_H+\gamma_{C_2HO_4^-} = \gamma_H+\gamma_{NO_3^-}$, and used the ISORROPIA-predicted
$\gamma_H+\gamma_{NO_3^-}$ value of 0.07. We used the average of $H_{C_2H_2O_4}$ values provided by Clegg et al. (1996),
Compernolle and Muller (2014) and Saxena and Hildemann (1996) (6.11 x 10$^8$ mole $L^{-1}$ atm$^{-1}$ at
25 °C), and accounted for the effect of temperature using equation 19 in Sander (2015).
Although $K_{a1}$ also depends on temperature, we used the $K_{a1}$ value at 25 °C (5.62 x 10$^{-2}$ mole $L^{-1}$,
(Haynes, 2014)) for all the oxalic acid S curve calculations since equations that compute $K_{a1}$
values for pure aqueous oxalic acid solutions at different temperatures are not available in the
literature. In addition, the temperatures observed in this study were close to 25 °C (study-average
temperature = 23.4 ± 4.0 °C).

Different S curves for $\varepsilon(C_2O_4^{2-})$ are calculated using 1-hour average values obtained from

the diurnal profiles of temperature and $W_i$ (specifically at 00:30, 06:30 and 12:30). The shape of
the S curve changes with the time of day due to the diurnal variations of temperature and $W_i$ (Fig
S11 and SI section S3). The S curves for $\varepsilon(C_2O_4^{2-})$ are very different from those of other acids,
such as $\varepsilon(NO_3^-)$ (shown in Fig. 4b). From the S curves for $\varepsilon(C_2O_4^{2-})$, which are calculated using
conditions in this study, some molar fraction of oxalic acid is always expected to be present in
the particle phase, even at low particle pH (i.e., the S curve does not go to zero at low pH). In
contrast, HNO$_3$ is expected to be present primarily in the gas phase at low particle pH (i.e., pH <
1) under similar temperature and $W_i$ conditions. This is due primarily to differences in the
Henry's law constants for the two acids. $H_{HNO_3}$ (2.57 x 10$^5$ mole $L^{-1}$ atm$^{-1}$) at 23.4 °C is three
orders of magnitude smaller than $H_{C_2H_2O_4}$ (7.27 x 10$^8$ mole $L^{-1}$ atm$^{-1}$) (Clegg and Brimblecombe,
1990; Sander, 2015). This means that some undissociated form of oxalate can be found in the



particle phase at any pH, and the molar fraction of this form increases with particle $W_i$ (see Fig.
S11). Oxalic acid's very high Henry's law constant combined with the $W_i$ conditions in this
study ensures that some fraction of the organic acid will be in the particle phase regardless the
particle pH.

Figure 7 compares the measured $\varepsilon(C_2O_4^{2-})$ vs. ISORROPIA-predicted $PM_1$ pH to the

analytically calculated S curves(s). The S curve is calculated based on the average temperature
and $W_i$ from 13 September to 6 October (23.4 ± 4.0 °C and 1.6 ± 1.7 μg m$^{-3}$, respectively). We
also calculated the "upper" and "lower" bounds of this S curve based on one standard deviation
from the average temperature and average $W_i$. Temperature = 27.4 °C and $W_i$ = 0.5 μg m$^{-3}$ are
used for calculations of the "lower" bound, while temperature = 19.4 °C and $W_i$ = 3.3 μg m$^{-3}$ are
used for calculations of the "upper" bound. For the ambient data, a range in $W_i$ (0.5 to 4 μg m$^{-3}$)
and temperature (15 to 31 °C) is chosen to be close to the analytical calculation. As shown in
Fig. 7, the measured $\varepsilon(C_2O_4^{2-})$ generally converged around the S curve calculated using the
average temperature and $W_i$ values. Although there is some scatter, the measured ratios are
mostly within the "upper" and "lower" bounds of the S curve.

We can also use the S curves for $\varepsilon(C_2O_4^{2-})$ in Fig. 7 to understand how high $NH_3$ events

at the site affect oxalic acid gas-particle partitioning. Here we define high $NH_3$ events as periods
where the $NH_3$ concentration was higher than 13.3 ppb (which is the study-averaged $NH_3$
concentration + 1 standard deviation). As discussed in section 3.3, the $PM_1$ pH during high $NH_3$
events is 2.5 ± 0.6, which is slightly higher than the study-averaged $PM_1$ pH of 2.2 ± 0.6. Based
on the S curve calculated using the average temperature and $W_i$ values, $\varepsilon(C_2O_4^{2-})$ increases from
81 % to 89 % when particle pH increases from 2.2 to 2.5. While this result indicates that high
$NH_3$ concentrations can raise the particle pH sufficiently such that it can promote gas-to-particle
partitioning of oxalic acid, this is not always the case. Specifically, increasing the particle pH
from -2 (or lower) to 1 will not result in a significant increase in $\varepsilon(C_2O_4^{2-})$. Therefore, whether or
not particle pH, and consequently oxalic acid gas-particle partitioning, is sensitive to $NH_3$
concentration depends strongly on particle pH values.

We also examined how well the analytically calculated S curve for $\varepsilon(C_2O_4^{2-})$ captures

diurnal variations of the measured $\varepsilon(C_2O_4^{2-})$. The ambient data is divided into two 12 hour sets





(08:00 to 19:59 and 20:00 to 07:59) based on the diurnal profile of solar irradiance. Two S
curves and their corresponding "upper" and "lower" bounds are calculated based on the average
temperature and $W_i$ of the two data sets, and are subsequently compared to the ambient data. As
shown in Fig. S12, the measured $\varepsilon(C_2O_4{}^{2-})$ in both data sets are generally consistent with
predicted values.
A number of inferences can be drawn from the overall good agreement between the
predicted and measured molar fractions of oxalic acid in the particle phase in Figs. 7 and S11.
Our assumptions regarding the activity coefficients, Henry's law constant and acid dissociation
constants used in the S curve calculations of $\varepsilon(C_2O_4{}^{2-})$ are reasonable for the conditions of this
study (or are at least self-consistent). S curves can be used to estimate activity coefficients based
on gas-particle partitioning data in cases where they are not available in the literature if the other
parameters are known. Analytically calculated S curves are a simple way of exploring how the
gas-particle partitioning of semi-volatile inorganic and organic compounds in the atmosphere are
affected by the compound's physicochemical properties (e.g., Henry's law constants and acid
dissociation constants), temperature, $W_i$ and pH. Overall, these results indicate that particle-
phase oxalate is in equilibrium with gas-phase oxalic acid, and that particle pH can influence
particle-phase oxalate concentrations. It also showed that particle-phase oxalate can be found
over a broad pH range, and that the presence of oxalate does not necessarily provide insights of
the particle pH. Because of its high Henry's law constant, particle-phase oxalate can be found in
aerosols even at extremely low pH values (i.e., the flat region in Fig. 7), although our data cannot
be used to test this since ambient particle pH values in this study are too high.
**3.5.2 Formic and acetic acids**
Similar comparisons between the predicted and measured $\varepsilon(HCOO^-)$ and $\varepsilon(CH_3CO_2{}^-)$ can
also be made. Derivation of the equations for S curves describing the dependence of formic and
acetic acid gas-particle partitioning (i.e., $\varepsilon(HCOO^-) = HCOO^- / (HCOOH + HCOO^-)$ and
$\varepsilon(CH_3CO_2{}^-) = CH_3CO_2{}^- / (CH_3CO_2H + CH_3CO_2{}^-)$, respectively) on particle pH are similar to that
of $HNO_3$ since they are monoprotic acids:
$$\varepsilon(HCOO^-) = \frac{H_{HCOOH}W_iRT\left(\frac{\gamma_H + \gamma_{HCOO^-}}{\gamma_{HCOOH}}10^{-pH} + K_{a1}\right) \times 0.987 \times 10^{-14}}{\gamma_H + \gamma_{HCOO^-}10^{-pH} + H_{HCOOH}W_iRT\left(\frac{\gamma_H + \gamma_{HCOO^-}}{\gamma_{HCOOH}}10^{-pH} + K_{a1}\right) \times 0.987 \times 10^{-14}} \quad (5)$$





$$\varepsilon(CH_3CO_2^-) = \frac{H_{CH_3CO_2H}W_iRT\left(\frac{\gamma_H + \gamma_{CH_3CO_2^-}}{\gamma_{CH_3CO_2H}}10^{-pH} + K_{a1}\right)\times 0.987 \times 10^{-14}}{\gamma_H + \gamma_{CH_3CO_2^-}10^{-pH} + H_{CH_3CO_2H}W_iRT\left(\frac{\gamma_H + \gamma_{CH_3CO_2^-}}{\gamma_{CH_3CO_2H}}10^{-pH} + K_{a1}\right)\times 0.987 \times 10^{-14}} \quad (6)$$

where $H_{HCOOH}$ and $H_{CH_3CO_2H}$ (mole L$^{-1}$ atm$^{-1}$) are the Henry's law constants for formic and acetic
acid, $K_{a1}$'s (mole L$^{-1}$) are the first acid dissociation constants, $R$ is the gas constant (8.314 m$^3$ Pa
K$^{-1}$ mol$^{-1}$), $T$ is temperature (K), and $\gamma_i$'s are activity coefficients. We used the web version of
AIOMFAC (www.aiomfac.caltech.edu) (Zuend et al., 2008; Zuend et al., 2011; Zuend et al.,
2012) to compute study-averaged $\gamma_{HCOOH}$ and $\gamma_{CH_3COOH}$ values of 0.334 and 2.150, respectively.
Similar to the case of oxalic acid, we assumed that $\gamma_{H^+}\gamma_{HCOO^-} = \gamma_{H^+}\gamma_{CH_3COO^-} = \gamma_{H^+}\gamma_{NO_3^-}$, and
used the ISORROPIA-predicted $\gamma_{H^+}\gamma_{NO_3^-}$ value of 0.07. Temperature-dependent $H_{HCOOH}$ and
$H_{CH_3CO_2H}$ values are obtained from Sander (2015) using the same methodology employed to
determine temperature-dependent $H_{C_2H_2O_4}$ values. We used $K_{a1}$ values at 25 °C (1.78 x 10$^{-4}$ mole
L$^{-1}$ for formic acid, and 1.75 x 10$^{-5}$ mole L$^{-1}$ for acetic acid (Haynes, 2014)) for the S curve
calculations.
S curves for ε(HCOO$^-$) and ε(CH$_3$CO$_2$$^-$) calculated based on temperature = 23.4 °C and
$W_i$ = 1.6 μg m$^{-3}$ can be seen in Fig. 8. Practically no formic or acetic acids are predicted to
partition to the particle phase (relative to oxalic acid) for the range of PM$_1$ pH calculated in this
study. This is due to significant differences in the Henry's law constants and acid dissociation
constants for the three organic acids. $H_{HCOOH}$ and $H_{CH_3CO_2H}$ (9540 and 5370 mole L$^{-1}$ atm$^{-1}$,
respectively) at 23.4 °C are substantially smaller than $H_{C_2H_2O_4}$ (7.27 x 10$^8$ mole L$^{-1}$ atm$^{-1}$)
(Sander, 2015). The $K_{a1}$ values for formic and acetic acids (1.78 x 10$^{-4}$ and 1.75 x 10$^{-5}$ mole L$^{-1}$,
respectively) are also considerably smaller than the $K_{a1}$ value for oxalic acid (5.62 x 10$^{-2}$ mole L$^{-1}$
$^{1}$) (Haynes, 2014). Note that $H_{HNO_3}$ is between that of $H_{C_2H_2O_4}$ and those of $H_{HCOOH}$ and
$H_{CH_3CO_2H}$ (compare Fig. 4b with Figs. 7 and 8).
As shown in Fig. 8, higher than expected levels of formate and acetate are observed in the
particle phase. This has also been reported in previous studies (Liu et al., 2012). Laboratory tests
showed that the disagreement cannot be explained by positive biases in the particle-phase
formate and acetate PILS-HPIC measurements resulting from less than 100 % gas removal by
the carbon denuder. The measured denuder efficiency for formic acid was ≥ 99.97% (SI section




S4). The possibility that formic and acetic acid dimers in the aqueous phase (Schrier et al., 1964;
Gilson et al., 1997; Chen et al., 2008) may result in higher than predicted molar fractions of
formate and acetate in the particle phase was explored, but also could not explain the observed
gas-particle partitioning of these acids (SI section S5). The disagreement could be due to
incorrect Henry's law constants for formic and acetic acids. However, the Henry's law constants
for formic and acetic acid would have to be $\sim 10^4$ times and $\sim 3 \times 10^5$ times larger than their
literature values, respectively, in order for their S curves to match our measured molar fractions
of formic and acetic acid in the particle phase. More research is needed to explain this
disagreement.
**4. Summary**

Gas- and particle-phase measurements were conducted in Yorkville, Georgia (a rural

field site) during fall 2016. The goal of the field study was to understand how $NH_3$ affects
particle acidity, and consequently SOA formation through the gas-particle partitioning of semi-
volatile inorganic and organic compounds. Since it is a rural site surrounded by forest,
agricultural land and CAFOs, this study provided an opportunity for ambient observations in an
area impacted by high local emissions of BVOCs and $NH_3$.

$NH_3$ concentrations measured by the $NH_3$-CIMS ranged from 0.7 to 39.0 ppb (study

average $8.1 \pm 5.2$ ppb), which were substantially higher than typical levels in the southeastern
U.S.. $PM_1$ inorganic chemical composition, gas-phase $HNO_3$ and $NH_3$ concentrations,
temperature and RH were used as model inputs in the ISORROPIA-II thermodynamic model to
calculate $PM_1$ pH and $W_i$. $PM_1$ pH ranged from 0.9 to 3.8, with a study-averaged pH of $2.2 \pm$
0.6. The measured and predicted $HNO_3$-$NO_3^-$ and $NH_3$-$NH_4^+$ gas-particle partitioning ratios were
in good agreement. The measured gas-phase organic acids were estimated to contribute 30 % of
the overall $WSOC_g$ on a carbon mass basis, whereas measured particle-phase organic acids
comprised 6 % of the total organic aerosol mass concentration and 4 % of the overall $WSOC_p$ on
a carbon mass basis. Formic and acetic acids were the most abundant gas-phase organic acids,
with study averages of $2.2 \pm 1.6$ and $1.9 \pm 1.3$ µg m$^{-3}$, respectively. Oxalate was the most
abundant particle-phase water-soluble organic acid anion, with a study average of $0.07 \pm 0.05$ µg
m$^{-3}$. Measured oxalic acid gas-particle partitioning ratios generally agreed with analytical
predictions, which were based on oxalic acid's physicochemical properties (specifically, its





701 Henry's law constants, acid dissociation constants and activity coefficients), temperature, $W_i$ and

702 particle pH. The partitioning of oxalic acid to the particle phase is highly sensitive to temperature

703 and $W_i$. In contrast, the partitioning of formic and acetic acids to the particle phase were higher

704 than predicted for reasons currently unknown.

705  Although past air regulations have resulted in decreased sulfate, nitrate and ammonium

706 aerosol mass concentrations across the U.S., our study suggests that the current limited

707 regulation of $NH_3$ emissions may result in some increase in the organic aerosol mass

708 concentration due to increased gas-to-particle partitioning of some organic acids. However, in

709 this study, the effect was small since the organic acids comprised a small fraction of the overall

710 organic aerosol mass.

711 **5. Acknowledgements**

712  The authors thank Eric Edgerton (Atmospheric Research and Analysis, Inc.) for

713 providing SEARCH network measurements and meteorological data.

714 **6. Funding**

715  This publication was developed under U.S. Environmental Protection Agency (EPA)

716 STAR Grant R835882 awarded to Georgia Institute of Technology. It has not been formally

717 reviewed by the EPA. The views expressed in this document are solely those of the authors and

718 do not necessarily reflect those of the EPA. EPA does not endorse any products or commercial

719 services mentioned in this publication.

720 **7. References**

721 Bertram, A. K., Martin, S. T., Hanna, S. J., Smith, M. L., Bodsworth, A., Chen, Q., Kuwata, M.,

722 Liu, A., You, Y., and Zorn, S. R.: Predicting the relative humidities of liquid-liquid phase

723 separation, efflorescence, and deliquescence of mixed particles of ammonium sulfate, organic

724 material, and water using the organic-to-sulfate mass ratio of the particle and the oxygen-to-

725 carbon elemental ratio of the organic component, Atmos. Chem. Phys., 11, 10995-11006,

726 10.5194/acp-11-10995-2011, 2011.





Blanchard, C. L., Hidy, G. M., Tanenbaum, S., and Edgerton, E. S.: NMOC, ozone, and organic
aerosol in the southeastern United States, 1999-2007: 3. Origins of organic aerosol in Atlanta,
Georgia,    and    surrounding    areas,    Atmospheric    Environment,    45,    1291-1302,
10.1016/j.atmosenv.2010.12.004, 2011.
Blanchard, C. L., Hidy, G. M., Tanenbaum, S., Edgerton, E. S., and Hartsell, B. E.: The
Southeastern Aerosol Research and Characterization (SEARCH) study: Temporal trends in gas
and PM concentrations and composition, 1999-2010, Journal of the Air & Waste Management
Association, 63, 247-259, 10.1080/10962247.2012.748523, 2013a.
Blanchard, C. L., Hidy, G. M., Tanenbaum, S., Edgerton, E. S., and Hartsell, B. E.: The
Southeastern Aerosol Research and Characterization (SEARCH) study: Spatial variations and
chemical climatology, 1999-2010, Journal of the Air & Waste Management Association, 63,
260-275, 10.1080/10962247.2012.749816, 2013b.
Blanchard, C. L., Tanenbaum, S., and Hidy, G. M.: Source Attribution of Air Pollutant
Concentrations and Trends in the Southeastern Aerosol Research and Characterization
(SEARCH)    Network,    Environmental    Science    &    Technology,    47,    13536-13545,
10.1021/es402876s, 2013c.
Bougiatioti, A., Nikolaou, P., Stavroulas, I., Kouvarakis, G., Weber, R., Nenes, A., Kanakidou,
M., and Mihalopoulos, N.: Particle water and pH in the eastern Mediterranean: source variability
and implications for nutrient availability, Atmos. Chem. Phys., 16, 4579-4591, 10.5194/acp-16-

746    4579-2016, 2016.

Canagaratna, M. R., Jayne, J. T., Jimenez, J. L., Allan, J. D., Alfarra, M. R., Zhang, Q., Onasch,
T. B., Drewnick, F., Coe, H., Middlebrook, A., Delia, A., Williams, L. R., Trimborn, A. M.,
Northway, M. J., DeCarlo, P. F., Kolb, C. E., Davidovits, P., and Worsnop, D. R.: Chemical and
microphysical characterization of ambient aerosols with the aerodyne aerosol mass spectrometer,
Mass Spectrometry Reviews, 26, 185-222, 10.1002/mas.20115, 2007.
Canagaratna, M. R., Jimenez, J. L., Kroll, J. H., Chen, Q., Kessler, S. H., Massoli, P.,
Hildebrandt Ruiz, L., Fortner, E., Williams, L. R., Wilson, K. R., Surratt, J. D., Donahue, N. M.,
Jayne, J. T., and Worsnop, D. R.: Elemental ratio measurements of organic compounds using



aerosol mass spectrometry: characterization, improved calibration, and implications, Atmos.
Chem. Phys., 15, 253-272, 10.5194/acp-15-253-2015, 2015.
Case, J. L., and Zavodsky, B. T.: Evolution of 2016 drought in the Southeastern United States
from a Land surface modeling perspective, Results in Physics, 8, 654-656,
10.1016/j.rinp.2017.12.029, 2018.
Chen, J. H., Brooks, C. L., and Scheraga, H. A.: Revisiting the carboxylic acid dimers in aqueous
solution: Interplay of hydrogen bonding, hydrophobic interactions, and entropy, Journal of
Physical Chemistry B, 112, 242-249, 10.1021/jp074355h, 2008.
Clegg, S. L., and Brimblecombe, P.: Equilibrium partial pressures and mean activity and osmotic
coefficients of 0-100-percent nitric- acid as a function of temperature, Journal of Physical
Chemistry, 94, 5369-5380, 10.1021/j100376a038, 1990.
Clegg, S. L., Brimblecombe, P., and Khan, L.: The Henry's law constant of oxalic acid and its
partitioning into the atmospheric aerosol, Idojaras, 100, 51-68, 1996.
Clegg, S. L., Brimblecombe, P., and Wexler, A. S.: Thermodynamic model of the system H+-
NH4+-SO42--NO3--H2O at tropospheric temperatures, Journal of Physical Chemistry A, 102,
2137-2154, 10.1021/jp973042r, 1998.
DeCarlo, P. F., Kimmel, J. R., Trimborn, A., Northway, M. J., Jayne, J. T., Aiken, A. C., Gonin,
M., Fuhrer, K., Horvath, T., Docherty, K. S., Worsnop, D. R., and Jimenez, J. L.: Field-
deployable, high-resolution, time-of-flight aerosol mass spectrometer, Analytical Chemistry, 78,
8281-8289, 10.1021/ac061249n, 2006.
Dentener, F. J., and Crutzen, P. J.: A 3-DIMENSIONAL MODEL OF THE GLOBAL
AMMONIA CYCLE, Journal of Atmospheric Chemistry, 19, 331-369, 10.1007/bf00694492,
777  1994.

Eatough, D. J., Wadsworth, A., Eatough, D. A., Crawford, J. W., Hansen, L. D., and Lewis, E.
A.: A multiple-system, multi-channel diffusion denuder sampler for the determination of fine-
particulate organic material in the atmosphere, Atmospheric Environment. Part A. General
Topics, 27, 1213-1219, 10.1016/0960-1686(93)90247-V, 1993.





Edgerton, E. S., Hartsell, B. E., Saylor, R. D., Jansen, J. J., Hansen, D. A., and Hidy, G. M.: The
southeastern aerosol research and characterization study: Part II. Filter-based measurements of
fine and coarse particulate matter mass and composition, Journal of the Air & Waste
Management Association, 55, 1527-1542, 2005.
Edgerton, E. S., Hartsell, B. E., Saylor, R. D., Jansen, J. J., Hansen, D. A., and Hidy, G. M.: The
Southeastern Aerosol Research and Characterization Study, part 3: Continuous measurements of
fine particulate matter mass and composition, Journal of the Air & Waste Management
Association, 56, 1325-1341, 10.1080/10473289.2006.10464585, 2006.
Edgerton, E. S., Saylor, R. D., Hartsell, B. E., Jansen, J. J., and Hansen, D. A.: Ammonia and
ammonium measurements from the southeastern United States, Atmospheric Environment, 41,
3339-3351, 10.1016/j.atmosenv.2006.12.034, 2007.
Ellis, R. A., Murphy, J. G., Markovic, M. Z., VandenBoer, T. C., Makar, P. A., Brook, J., and
Mihele, C.: The influence of gas-particle partitioning and surface-atmosphere exchange on
ammonia during BAQS-Met, Atmos. Chem. Phys., 11, 133-145, 10.5194/acp-11-133-2011,

796    2011.

Ellis, R. A., Jacob, D. J., Sulprizio, M. P., Zhang, L., Holmes, C. D., Schichtel, B. A., Blett, T.,
Porter, E., Pardo, L. H., and Lynch, J. A.: Present and future nitrogen deposition to national
parks in the United States: critical load exceedances, Atmos. Chem. Phys., 13, 9083-9095,
10.5194/acp-13-9083-2013, 2013.
Ervens, B., Feingold, G., Frost, G. J., and Kreidenweis, S. M.: A modeling study of aqueous
production of dicarboxylic acids: 1. Chemical pathways and speciated organic mass production,
Journal of Geophysical Research-Atmospheres, 109, 10.1029/2003jd004387, 2004.
Fountoukis, C., and Nenes, A.: ISORROPIA II: a computationally efficient thermodynamic
equilibrium model for K+-Ca2+-Mg2+-Nh(4)(+)-Na+-SO42--NO3--Cl--H2O aerosols, Atmos.
Chem. Phys., 7, 4639-4659, 2007.





Gilson, M. K., Given, J. A., Bush, B. L., and McCammon, J. A.: The statistical-thermodynamic
basis for computation of binding affinities: A critical review, Biophysical Journal, 72, 1047-
1069, 10.1016/s0006-3495(97)78756-3, 1997.
Guenther, A. B., Jiang, X., Heald, C. L., Sakulyanontvittaya, T., Duhl, T., Emmons, L. K., and
Wang, X.: The Model of Emissions of Gases and Aerosols from Nature version 2.1
(MEGAN2.1): an extended and updated framework for modeling biogenic emissions,
Geoscientific Model Development, 5, 1471-1492, 10.5194/gmd-5-1471-2012, 2012.
Guo, H., Xu, L., Bougiatioti, A., Cerully, K. M., Capps, S. L., Hite, J. R., Jr., Carlton, A. G., Lee,
S. H., Bergin, M. H., Ng, N. L., Nenes, A., and Weber, R. J.: Fine-particle water and pH in the
southeastern United States, Atmos. Chem. Phys., 15, 5211-5228, 10.5194/acp-15-5211-2015,

817    2015.

Guo, H., Sullivan, A. P., Campuzano-Jost, P., Schroder, J. C., Lopez-Hilfiker, F. D., Dibb, J. E.,
Jimenez, J. L., Thornton, J. A., Brown, S. S., Nenes, A., and Weber, R. J.: Fine particle pH and
the partitioning of nitric acid during winter in the northeastern United States, Journal of
Geophysical Research-Atmospheres, 121, 10355-10376, 10.1002/2016jd025311, 2016.
Guo, H., Liu, J. M., Froyd, K. D., Roberts, J. M., Veres, P. R., Hayes, P. L., Jimenez, J. L.,
Nenes, A., and Weber, R. J.: Fine particle pH and gas-particle phase partitioning of inorganic
species in Pasadena, California, during the 2010 CalNex campaign, Atmos. Chem. Phys., 17,
5703-5719, 10.5194/acp-17-5703-2017, 2017a.
Guo, H., Nenes, A., and Weber, R. J.: The underappreciated role of nonvolatile cations on
aerosol ammonium-sulfate molar ratios, Atmos. Chem. Phys. Discuss., 2017, 1-19, 10.5194/acp-
2017-737, 2017b.
Guo, H., Weber, R. J., and Nenes, A.: High levels of ammonia do not raise fine particle pH
sufficiently to yield nitrogen oxide-dominated sulfate production, Scientific Reports, 7,
10.1038/s41598-017-11704-0, 2017c.



Hansen, D. A., Edgerton, E. S., Hartsell, B. E., Jansen, J. J., Kandasamy, N., Hidy, G. M., and
Blanchard, C. L.: The southeastern aerosol research and characterization study: Part 1-overview,
Journal of the Air & Waste Management Association, 53, 1460-1471, 2003.
Haynes, W. M.: CRC handbook of chemistry and physics: A ready-reference book of chemical
and physical data. , Boca Raton: CRC Press, 2014.
Hennigan, C. J., Izumi, J., Sullivan, A. P., Weber, R. J., and Nenes, A.: A critical evaluation of
proxy methods used to estimate the acidity of atmospheric particles, Atmos. Chem. Phys., 15,
2775-2790, 10.5194/acp-15-2775-2015, 2015.
Huey, L. G., Hanson, D. R., and Howard, C. J.: Reactions of SF6- and I- with Atmospheric Trace
Gases, Journal of Physical Chemistry, 99, 5001-5008, 10.1021/j100014a021, 1995.
Huey, L. G., Tanner, D. J., Slusher, D. L., Dibb, J. E., Arimoto, R., Chen, G., Davis, D., Buhr,
M. P., Nowak, J. B., Mauldin, R. L., Eisele, F. L., and Kosciuch, E.: CIMS measurements of
HNO3 and SO2 at the South Pole during ISCAT 2000, Atmospheric Environment, 38, 5411-
5421, 10.1016/j.atmosenv.2004.04.037, 2004.
Kawamura, K., and Gagosian, R. B.: Implication of omega-oxocarboxylic acids in the remote
marine atmosphere for photo-oxidation of unsaturated fatty acids, Nature, 325, 330-332,
10.1038/325330a0, 1987.
Kawamura, K., and Ikushima, K.: Seasonal changes in the distribution of dicarboxylic acids in
the urban atmosphere, Environmental Science & Technology, 27, 2227-2235,
10.1021/es00047a033, 1993.
Kawamura, K., and Bikkina, S.: A review of dicarboxylic acids and related compounds in
atmospheric aerosols: Molecular distributions, sources and transformation, Atmospheric
Research, 170, 140-160, 10.1016/j.atmosres.2015.11.018, 2016.
Kerminen, V. M., Ojanen, C., Pakkanen, T., Hillamo, R., Aurela, M., and Merilainen, J.: Low-
molecular-weight dicarboxylic acids in an urban and rural atmosphere, Journal of Aerosol
Science, 31, 349-362, 10.1016/s0021-8502(99)00063-4, 2000.





Laskin, A., Laskin, J., and Nizkorodov, S. A.: Chemistry of Atmospheric Brown Carbon,
Chemical Reviews, 115, 4335-4382, 10.1021/cr5006167, 2015.
Laskin, J., Laskin, A., Roach, P. J., Slysz, G. W., Anderson, G. A., Nizkorodov, S. A., Bones, D.
L., and Nguyen, L. Q.: High-Resolution Desorption Electrospray Ionization Mass Spectrometry
for Chemical Characterization of Organic Aerosols, Analytical Chemistry, 82, 2048-2058,
10.1021/ac902801f, 2010.
Lee, H. J., Laskin, A., Laskin, J., and Nizkorodov, S. A.: Excitation-Emission Spectra and
Fluorescence Quantum Yields for Fresh and Aged Biogenic Secondary Organic Aerosols,
Environmental Science & Technology, 47, 5763-5770, 10.1021/es400644c, 2013.
Lim, H. J., Carlton, A. G., and Turpin, B. J.: Isoprene forms secondary organic aerosol through
cloud processing: Model simulations, Environmental Science & Technology, 39, 4441-4446,
10.1021/es048039h, 2005.
Liu, J., Zhang, X., Parker, E. T., Veres, P. R., Roberts, J. M., de Gouw, J. A., Hayes, P. L.,
Jimenez, J. L., Murphy, J. G., Ellis, R. A., Huey, L. G., and Weber, R. J.: On the gas-particle
partitioning of soluble organic aerosol in two urban atmospheres with contrasting emissions: 2.
Gas and particle phase formic acid, Journal of Geophysical Research-Atmospheres, 117,
10.1029/2012jd017912, 2012.
Malm, W. C., and Day, D. E.: Estimates of aerosol species scattering characteristics as a function
of relative humidity, Atmospheric Environment, 35, 2845-2860, 10.1016/s1352-2310(01)00077-
877    2, 2001.

Middlebrook, A. M., Bahreini, R., Jimenez, J. L., and Canagaratna, M. R.: Evaluation of
Composition-Dependent Collection Efficiencies for the Aerodyne Aerosol Mass Spectrometer
using Field Data, Aerosol Science and Technology, 46, 258-271,
881    10.1080/02786826.2011.620041, 2012.

Na, K., Song, C., Switzer, C., and Cocker, D. R.: Effect of ammonia on secondary organic
aerosol formation from alpha-Pinene ozonolysis in dry and humid conditions, Environmental
Science & Technology, 41, 6096-6102, 10.1021/es061956y, 2007.





Nah, T., Ji, Y., Tanner, D. J., Guo, H., Sullivan, A. P., Ng, N. L., Weber, R. J., and Huey, L. G.:
Real-time measurements of gas-phase organic acids using SF6- chemical ionization mass
spectrometry, Atmos. Meas. Tech. Discuss., 2018, 1-40, 10.5194/amt-2018-46, 2018.
Nenes, A., Pandis, S. N., and Pilinis, C.: ISORROPIA: A new thermodynamic equilibrium model
for multiphase multicomponent inorganic aerosols, Aquatic Geochemistry, 4, 123-152,
10.1023/a:1009604003981, 1998.
Neuman, J. A., Ryerson, T. B., Huey, L. G., Jakoubek, R., Nowak, J. B., Simons, C., and
Fehsenfeld, F. C.: Calibration and evaluation of nitric acid and ammonia permeation tubes by
UV optical absorption, Environmental Science & Technology, 37, 2975-2981,
10.1021/es0264221, 2003.
Nowak, J. B., Huey, L. G., Eisele, F. L., Tanner, D. J., Mauldin, R. L., Cantrell, C., Kosciuch, E.,
and Davis, D. D.: Chemical ionization mass spectrometry technique for detection of
dimethylsulfoxide and ammonia, Journal of Geophysical Research-Atmospheres, 107,
10.1029/2001jd001058, 2002.
Nowak, J. B., Huey, L. G., Russell, A. G., Tian, D., Neuman, J. A., Orsini, D., Sjostedt, S. J.,
Sullivan, A. P., Tanner, D. J., Weber, R. J., Nenes, A., Edgerton, E., and Fehsenfeld, F. C.:
Analysis of urban gas phase ammonia measurements from the 2002 Atlanta Aerosol Nucleation
and Real-Time Characterization Experiment (ANARChE), Journal of Geophysical Research-
Atmospheres, 111, 14, 10.1029/2006jd007113, 2006.
Orsini, D. A., Ma, Y. L., Sullivan, A., Sierau, B., Baumann, K., and Weber, R. J.: Refinements to
the particle-into-liquid sampler (PILS) for ground and airborne measurements of water soluble
aerosol composition, Atmospheric Environment, 37, 1243-1259, 10.1016/s1352-2310(02)01015-

907    4, 2003.

Park Williams, A., Cook, B. I., Smerdon, J. E., Bishop, D. A., Seager, R., and Mankin, J. S.: The
2016 Southeastern U.S. Drought: An Extreme Departure From Centennial Wetting and Cooling,
Journal of Geophysical Research: Atmospheres, 122, 10,888-810,905, 10.1002/2017JD027523,

911    2017.




Pechony, O., and Shindell, D. T.: Driving forces of global wildfires over the past millennium and
the forthcoming century, Proceedings of the National Academy of Sciences of the United States
of America, 107, 19167-19170, 10.1073/pnas.1003669107, 2010.
Reis, S., Pinder, R. W., Zhang, M., Lijie, G., and Sutton, M. A.: Reactive nitrogen in
atmospheric emission inventories, Atmos. Chem. Phys., 9, 7657-7677, 10.5194/acp-9-7657-
917   2009, 2009.

Sander, R.: Compilation of Henry's law constants (version 4.0) for water as solvent, Atmos.
Chem. Phys., 15, 4399-4981, 10.5194/acp-15-4399-2015, 2015.
Saxena, P., and Hildemann, L. M.: Water-soluble organics in atmospheric particles: A critical
review of the literature and application of thermodynamics to identify candidate compounds,
Journal of Atmospheric Chemistry, 24, 57-109, 10.1007/bf00053823, 1996.
Saylor, R., Myles, L., Sibble, D., Caldwell, J., and Xing, J.: Recent trends in gas-phase ammonia
and PM2.5 ammonium in the Southeast United States, Journal of the Air & Waste Management
Association, 65, 347-357, 10.1080/10962247.2014.992554, 2015.
Schrier, E. E., Pottle, M., and Scheraga, H. A.: The Influence of Hydrogen and Hydrophobic
Bonds on the Stability of the Carboxylic Acid Dimers in Aqueous Solution, Journal of the
American Chemical Society, 86, 3444-3449, 10.1021/ja01071a009, 1964.
Seinfeld, J. H., and Pandis, S. N.: Atmospheric chemistry and physics : from air pollution to
climate change, Third edition. ed., John Wiley & Sons, Inc., Hoboken, New Jersey, xxvi, 1120
pages pp., 2016.
Shi, G. L., Xu, J., Peng, X., Xiao, Z. M., Chen, K., Tian, Y. Z., Guan, X. B., Feng, Y. C., Yu, H.
F., Nenes, A., and Russell, A. G.: pH of Aerosols in a Polluted Atmosphere: Source
Contributions to Highly Acidic Aerosol, Environmental Science & Technology, 51, 4289-4296,
10.1021/acs.est.6b05736, 2017.
Song, M., Marcolli, C., Krieger, U. K., Zuend, A., and Peter, T.: Liquid-liquid phase separation
and morphology of internally mixed dicarboxylic acids/ammonium sulfate/water particles,
Atmos. Chem. Phys., 12, 2691-2712, 10.5194/acp-12-2691-2012, 2012.





Song, S., Gao, M., Xu, W., Shao, J., Shi, G., Wang, S., Wang, Y., Sun, Y., and McElroy, M. B.:
Fine particle pH for Beijing winter haze as inferred from different thermodynamic equilibrium
models, Atmos. Chem. Phys. Discuss., 2018, 1-26, 10.5194/acp-2018-6, 2018.
Sorooshian, A., Ng, N. L., Chan, A. W. H., Feingold, G., Flagan, R. C., and Seinfeld, J. H.:
Particulate organic acids and overall water-soluble aerosol composition measurements from the
2006 Gulf of Mexico Atmospheric Composition and Climate Study (GoMACCS), Journal of
Geophysical Research-Atmospheres, 112, 16, 10.1029/2007jd008537, 2007.
Sorooshian, A., Murphy, S. M., Hersey, S., Bahreini, R., Jonsson, H., Flagan, R. C., and
Seinfeld, J. H.: Constraining the contribution of organic acids and AMS m/z 44 to the organic
aerosol budget: On the importance of meteorology, aerosol hygroscopicity, and region,
Geophysical Research Letters, 37, 5, 10.1029/2010gl044951, 2010.
Spaulding, R. S., Talbot, R. W., and Charles, M. J.: Optimization of a mist chamber (cofer
scrubber) for sampling water-soluble organics in air, Environmental Science & Technology, 36,
1798-1808, 10.1021/es011189x, 2002.
Sullivan, A. P., Weber, R. J., Clements, A. L., Turner, J. R., Bae, M. S., and Schauer, J. J.: A
method for on-line measurement of water-soluble organic carbon in ambient aerosol particles:
Results from an urban site, Geophysical Research Letters, 31, 10.1029/2004gl019681, 2004.
Sun, K., Tao, L., Miller, D. J., Pan, D., Golston, L. M., Zondlo, M. A., Griffin, R. J., Wallace, H.
W., Leong, Y. J., Yang, M. M., Zhang, Y., Mauzerall, D. L., and Zhu, T.: Vehicle Emissions as
an Important Urban Ammonia Source in the United States and China, Environmental Science &
Technology, 51, 2472-2481, 10.1021/acs.est.6b02805, 2017.
Updyke, K. M., Nguyen, T. B., and Nizkorodov, S. A.: Formation of brown carbon via reactions
of ammonia with secondary organic aerosols from biogenic and anthropogenic precursors,
Atmospheric Environment, 63, 22-31, 10.1016/j.atmosenv.2012.09.012, 2012.
Van Damme, M., Clarisse, L., Heald, C. L., Hurtmans, D., Ngadi, Y., Clerbaux, C., Dolman, A.
J., Erisman, J. W., and Coheur, P. F.: Global distributions, time series and error characterization





of atmospheric ammonia (NH3) from IASI satellite observations, Atmos. Chem. Phys., 14, 2905-
2922, 10.5194/acp-14-2905-2014, 2014.
Warner, J. X., Wei, Z. G., Strow, L. L., Dickerson, R. R., and Nowak, J. B.: The global
tropospheric ammonia distribution as seen in the 13-year AIRS measurement record, Atmos.
Chem. Phys., 16, 5467-5479, 10.5194/acp-16-5467-2016, 2016.
Warner, J. X., Dickerson, R. R., Wei, Z., Strow, L. L., Wang, Y., and Liang, Q.: Increased
atmospheric ammonia over the world's major agricultural areas detected from space, Geophysical
Research Letters, 44, 2875-2884, 10.1002/2016gl072305, 2017.
Weber, R. J., Orsini, D., Daun, Y., Lee, Y. N., Klotz, P. J., and Brechtel, F.: A particle-into-
liquid collector for rapid measurement of aerosol bulk chemical composition, Aerosol Science
and Technology, 35, 718-727, 10.1080/02786820152546761, 2001.
Weber, R. J., Lee, S., Chen, G., Wang, B., Kapustin, V., Moore, K., Clarke, A. D., Mauldin, L.,
Kosciuch, E., Cantrell, C., Eisele, F., Thornton, D. C., Bandy, A. R., Sachse, G. W., and
Fuelberg, H. E.: New particle formation in anthropogenic plumes advecting from Asia observed
during    TRACE-P,    Journal    of    Geophysical    Research-Atmospheres,    108,    13,
10.1029/2002jd003112, 2003.
Weber, R. J., Guo, H., Russell, A. G., and Nenes, A.: High aerosol acidity despite declining
atmospheric sulfate concentrations over the past 15 years, Nature Geoscience, 9, 282-+,
10.1038/ngeo2665, 2016.
Xing, J., Pleim, J., Mathur, R., Pouliot, G., Hogrefe, C., Gan, C. M., and Wei, C.: Historical
gaseous and primary aerosol emissions in the United States from 1990 to 2010, Atmos. Chem.
Phys., 13, 7531-7549, 10.5194/acp-13-7531-2013, 2013.
Xu, L., Guo, H., Boyd, C. M., Klein, M., Bougiatioti, A., Cerully, K. M., Hite, J. R., Isaacman-
VanWertz, G., Kreisberg, N. M., Knote, C., Olson, K., Koss, A., Goldstein, A. H., Hering, S. V.,
de Gouw, J., Baumann, K., Lee, S.-H., Nenes, A., Weber, R. J., and Ng, N. L.: Effects of
anthropogenic   emissions   on   aerosol   formation   from   isoprene   and   monoterpenes   in   the





southeastern United States, Proceedings of the National Academy of Sciences of the United
States of America, 112, 37-42, 10.1073/pnas.1417609112, 2015a.
Xu, L., Suresh, S., Guo, H., Weber, R. J., and Ng, N. L.: Aerosol characterization over the
southeastern United States using high-resolution aerosol mass spectrometry: spatial and seasonal
variation of aerosol composition and sources with a focus on organic nitrates, Atmos. Chem.
Phys., 15, 7307-7336, 10.5194/acp-15-7307-2015, 2015b.
Xu, L., Guo, H. Y., Weber, R. J., and Ng, N. L.: Chemical Characterization of Water-Soluble
Organic Aerosol in Contrasting Rural and Urban Environments in the Southeastern United
States, Environmental Science & Technology, 51, 78-88, 10.1021/acs.est.6b05002, 2017.
Yao, X. H., Hu, Q. J., Zhang, L. M., Evans, G. J., Godri, K. J., and Ng, A. C.: Is vehicular
emission a significant contributor to ammonia in the urban atmosphere?, Atmospheric
Environment, 80, 499-506, 10.1016/j.atmosenv.2013.08.028, 2013.
You, Y., Renbaum-Wolff, L., and Bertram, A. K.: Liquid-liquid phase separation in particles
containing organics mixed with ammonium sulfate, ammonium bisulfate, ammonium nitrate or
sodium chloride, Atmos. Chem. Phys., 13, 11723-11734, 10.5194/acp-13-11723-2013, 2013.
You, Y., Kanawade, V. P., de Gouw, J. A., Guenther, A. B., Madronich, S., Sierra-Hernandez,
M. R., Lawler, M., Smith, J. N., Takahama, S., Ruggeri, G., Koss, A., Olson, K., Baumann, K.,
Weber, R. J., Nenes, A., Guo, H., Edgerton, E. S., Porcelli, L., Brune, W. H., Goldstein, A. H.,
and Lee, S. H.: Atmospheric amines and ammonia measured with a chemical ionization mass
spectrometer (CIMS), Atmos. Chem. Phys., 14, 12181-12194, 10.5194/acp-14-12181-2014,
2014a.
You, Y., Smith, M. L., Song, M., Martin, S. T., and Bertram, A. K.: Liquid-liquid phase
separation in atmospherically relevant particles consisting of organic species and inorganic salts,
International Reviews in Physical Chemistry, 33, 43-77, 10.1080/0144235x.2014.890786, 2014b.
You, Y., and Bertram, A. K.: Effects of molecular weight and temperature on liquid-liquid phase
separation in particles containing organic species and inorganic salts, Atmos. Chem. Phys., 15,
1351-1365, 10.5194/acp-15-1351-2015, 2015.





Yu, H., and Lee, S. H.: Chemical ionisation mass spectrometry for the measurement of
atmospheric amines, Environ. Chem., 9, 190-201, 10.1071/en12020, 2012.
Zuend, A., Marcolli, C., Luo, B. P., and Peter, T.: A thermodynamic model of mixed organic-
inorganic aerosols to predict activity coefficients, Atmos. Chem. Phys., 8, 4559-4593,
10.5194/acp-8-4559-2008, 2008.
Zuend, A., Marcolli, C., Booth, A. M., Lienhard, D. M., Soonsin, V., Krieger, U. K., Topping, D.
O., McFiggans, G., Peter, T., and Seinfeld, J. H.: New and extended parameterization of the
thermodynamic model AIOMFAC: calculation of activity coefficients for organic-inorganic
mixtures containing carboxyl, hydroxyl, carbonyl, ether, ester, alkenyl, alkyl, and aromatic
functional groups, Atmos. Chem. Phys., 11, 9155-9206, 10.5194/acp-11-9155-2011, 2011.
Zuend, A., Marcolli, C., Luo, B. P., and Peter, T.: A thermodynamic model of mixed organic-
inorganic aerosols to predict activity coefficients (vol 8, pg 4559, 2008), Atmos. Chem. Phys.,
12, 10075-10075, 10.5194/acp-12-10075-2012, 2012.











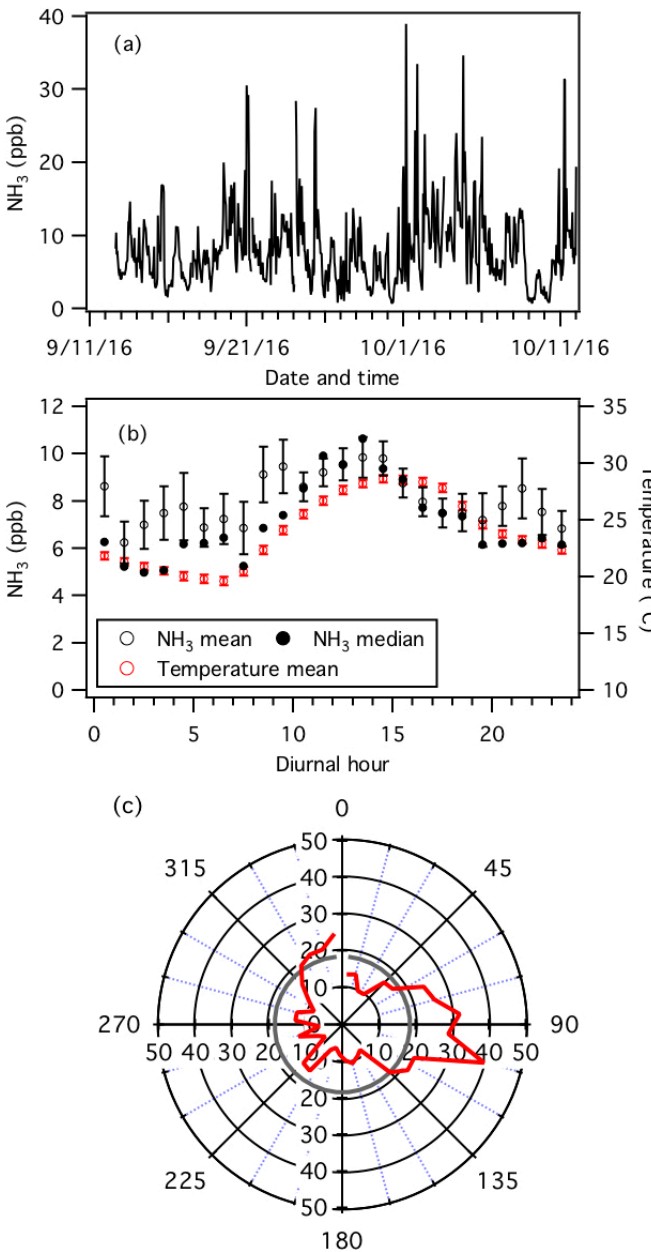


**Figure 1:** Measurements by the NH₃-CIMS during the second half of the study. (a) Time series
of NH₃ concentration. The data is displayed as 1-hour averages. (b) Diurnal profiles of NH₃
concentration (mean and median) and temperature. Error bars shown are the standard errors.
Dates and times displayed are local time. All the concentrations represent averages in 1-hour





intervals and the standard errors are plotted as error bars. (c) Average NH$_3$ concentration
normalized to wind speed (i.e., NH$_3$ concentration (ppb) x wind speed (m s$^{-1}$)) in each 10 degrees
bin (red line). The study-averaged normalized NH$_3$ concentration is shown as a grey line.

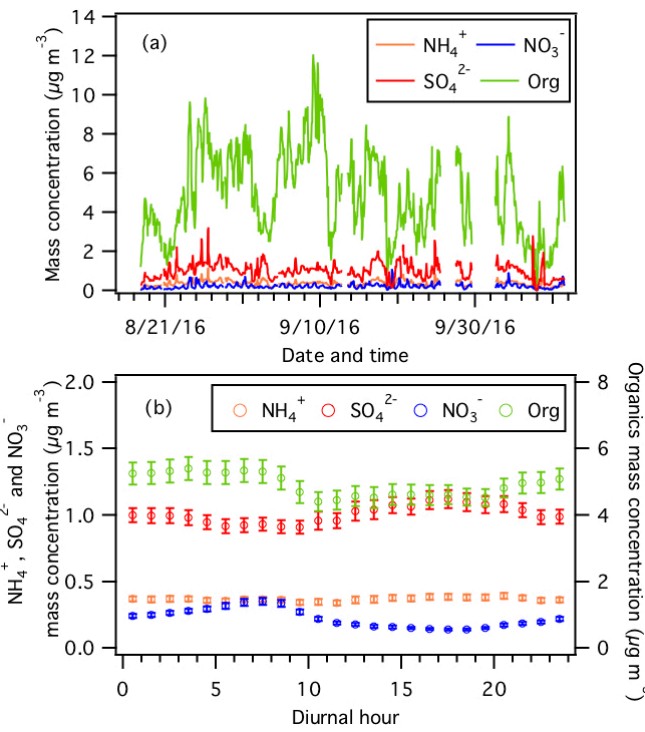


**Figure 2:** (a) Time series and (b) diurnal profiles of non-refractory PM$_1$ species measured by the
AMS. Error bars shown in panel (b) are the standard errors. Dates and times displayed are local
time. All the mass concentrations shown here are obtained from scaling the raw data by 0.5.
Refer to the text for details.





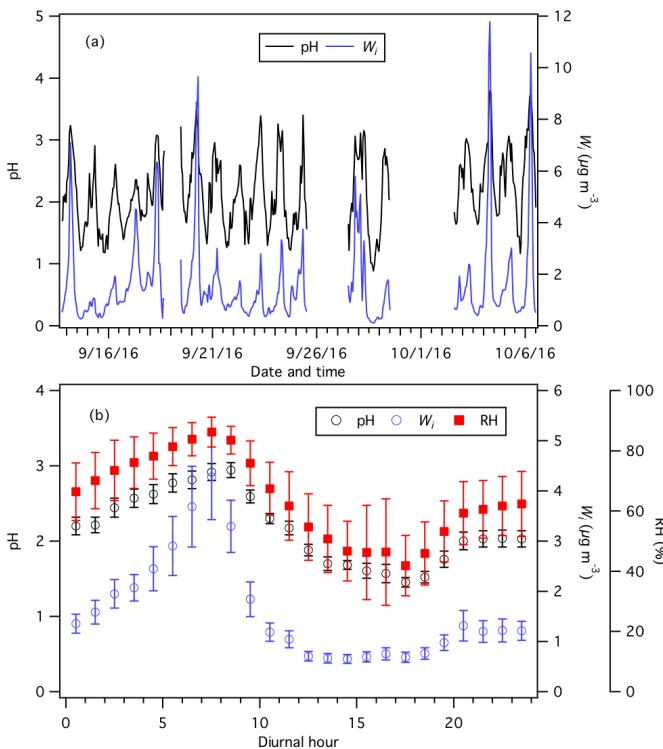


**Figure 3:** (a) Time series and (b) diurnal profiles of ISORROPIA-predicted $PM_1$ pH and $W_i$.
Dates and times displayed are local time. All the data shown here represent averages in 1-hour
intervals. Error bars shown in panel (b) are the standard errors.



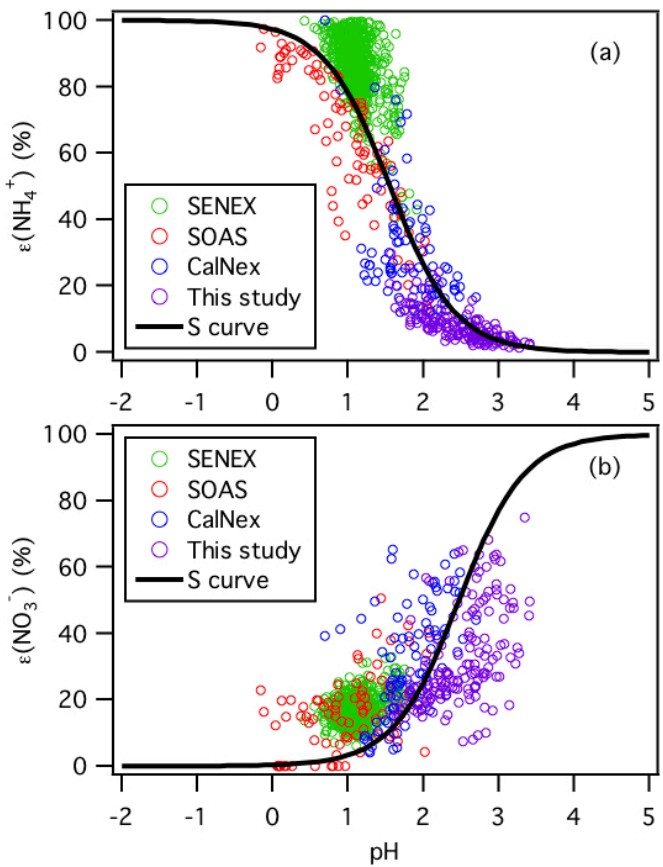


**Figure 4:** Analytically calculated S curves of $\varepsilon(NH_4^+)$ and $\varepsilon(NO_3^-)$ and ambient data plotted against ISORROPIA-predicted particle pH for this study, SENEX, SOAS and CalNex. For the ambient datasets, a narrow range of $W_i$ (1 to 4 µg m$^{-3}$) and temperature (15 to 25 °C) are selected to be close to the analytical calculation input (i.e., $W_i$ = 2.5 µg m$^{-3}$ and temperature = 20 °C). Similar to Guo et al. (2017a), $\gamma_{NH_4^+} = 1$ and $\gamma_{H^+-NO_3^-} = \sqrt{\gamma_{H^+}\gamma_{NO_3^-}} = 0.28$ are used for the analytically calculated S curves.






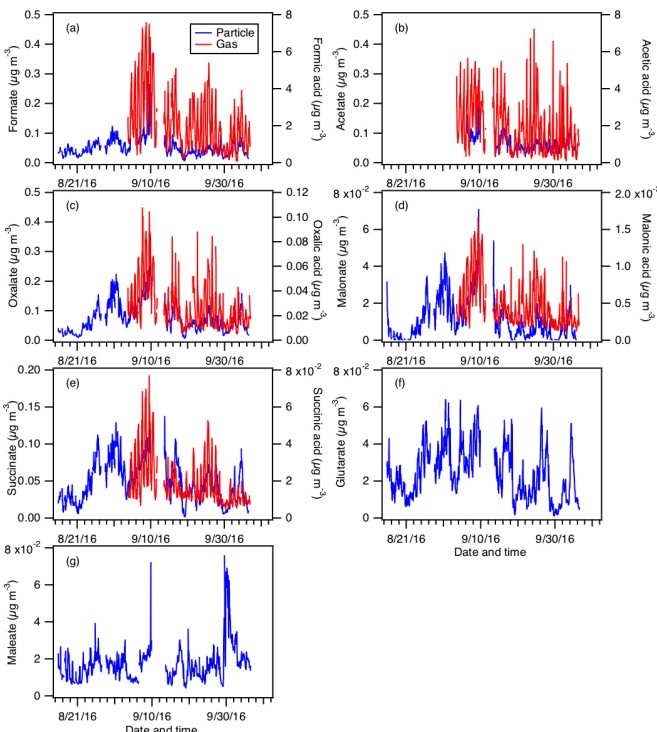

**Figure 5:** Particle- and gas-phase measurements of (a) formic, (b) acetic, (c) oxalic, (d) malonic,
(e) succinic, (f) glutaric, and (g) maleic acids. Particle-phase measurements are shown on the left
y axes, while gas-phase measurements are shown on the right y axes. Dates and times displayed
are local time. Gas-phase measurements of glutaric and maleic acids are not available.





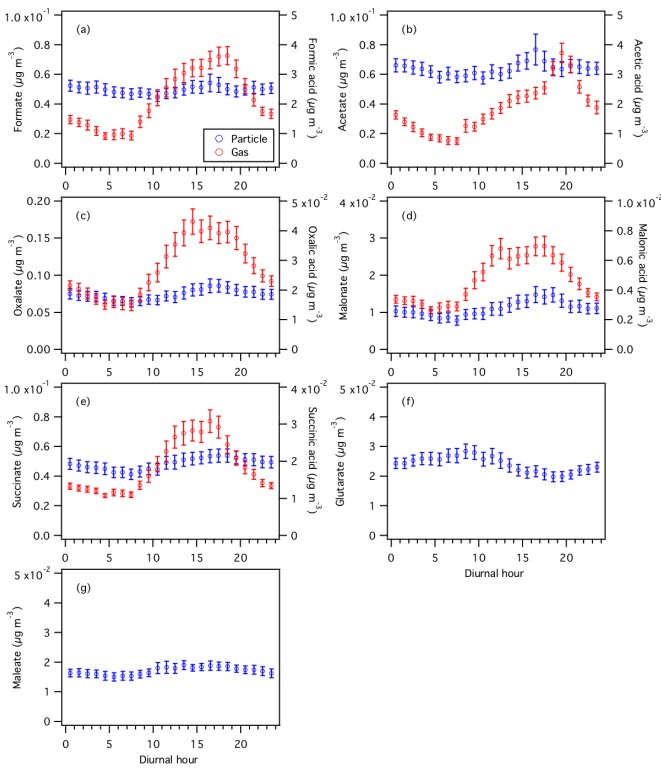


**Figure 6:** Diurnal profiles of particle- and gas-phase (a) formic, (b) acetic, (c) oxalic, (d) malonic, (e) succinic, (f) glutaric, and (g) maleic acids. Particle-phase measurements are shown on the left y axes, while gas-phase measurements are shown on the right y axes. All the data shown here represent averages in 1-hour intervals. Error bars shown are the standard errors.

1077

1078





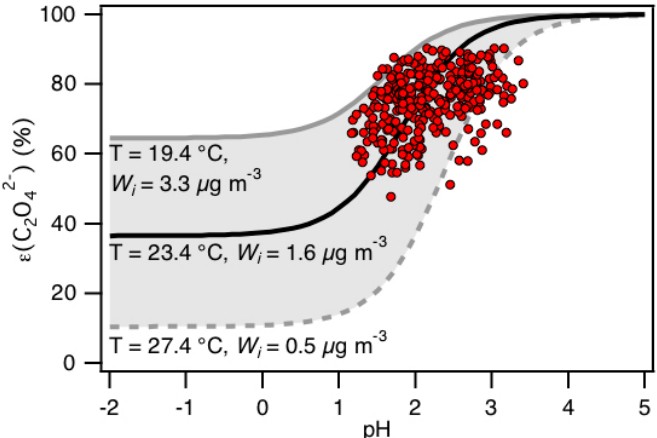

1079

**Figure 7:** Analytically calculated S curve of $\varepsilon(C_2O_4^{2-})$ and ambient data from 13 September to 6 October 2016 plotted against ISORROPIA-predicted particle pH. For the ambient data, a range in $W_i$ (0.5 to 4 $\mu$g m$^{-3}$) and temperature (15 to 31 °C) are chosen to be close to the analytically calculated outputs. For the analytically calculated S curves, we used $\gamma_{C_2H_2O_4}$ = 0.0492 (AIOMFAC predicted). We also assumed that $\gamma_{H^+}\gamma_{C_2HO_4^-}$ = $\gamma_{H^+}\gamma_{NO_3^-}$, and used the ISORROPIA-predicted $\gamma_{H^+-NO_3^-}$ = $\sqrt{\gamma_{H^+}\gamma_{NO_3^-}}$ = 0.265. The black line is the S curve calculated using the selected time period's average temperature (23.4 ± 4.0 °C) and $W_i$ (1.6 ± 1.7 $\mu$g m$^{-3}$). The grey lines are S curves calculated using one standard deviation from the average temperature and $W_i$ (i.e., temperature = 27.4 °C and $W_i$ = 0.5 $\mu$g m$^{-3}$ for dotted grey line, temperature = 19.4 °C and $W_i$ = 3.3 $\mu$g m$^{-3}$ for solid grey line).



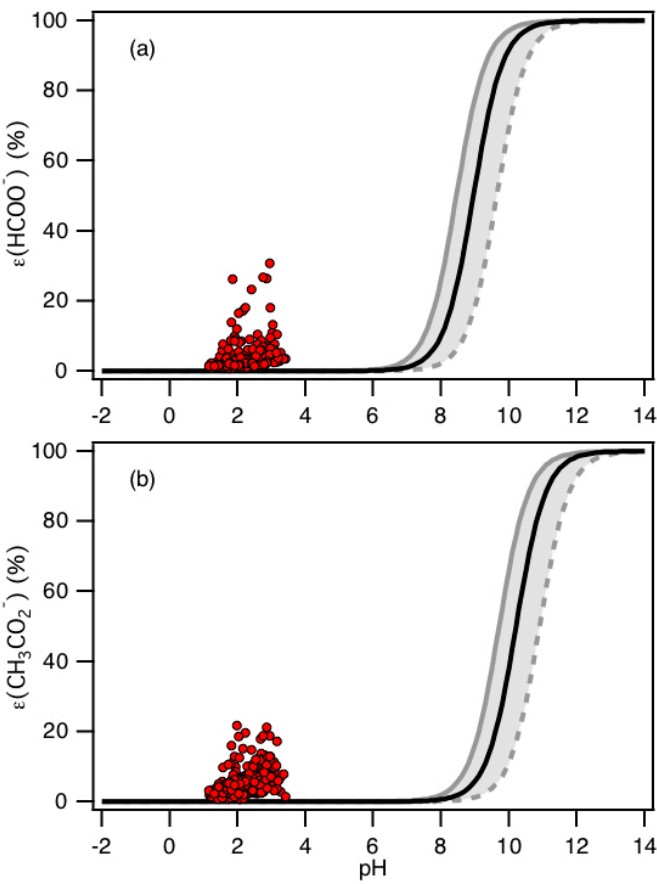

1090

**Figure 8:** Analytically calculated S curves of ε(HCOO⁻) and ε(CH₃CO₂⁻) (solid black lines) and ambient data from 13 September to 6 October 2016 plotted against ISORROPIA-predicted particle pH (shown in panels (a) and (b), respectively). For the ambient data, a narrow range in $W_i$ (0.5 to 4 µg m⁻³) and RH (20 to 90 %) is chosen to be close to the analytically calculated outputs. For the analytically calculated S curves, we used $\gamma_{HCOOH} = 0.334$ and $\gamma_{CH_3COOH} = 2.150$ (AIOMFAC predicted). We also assumed that $\gamma_{H^+}\gamma_{HCOO^-} = \gamma_{H^+}\gamma_{CH_3COO^-} = \gamma_{H^+}\gamma_{NO_3^-}$, and used the ISORROPIA-predicted $\gamma_{H^+-NO_3^-} = \sqrt{\gamma_{H^+}\gamma_{NO_3^-}} = 0.265$. The black lines are S curves calculated using the selected time period's average temperature (23.4 ± 4.0 °C) and $W_i$ (1.6 ± 1.7 µg m⁻³). The grey lines are S curves calculated using one standard deviation from the average temperature and $W_i$ (i.e., temperature = 27.4 °C and $W_i$ = 0.5 µg m⁻³ for dotted grey line, temperature = 19.4 °C and $W_i$ = 3.3 µg m⁻³ for solid grey line).





**Table 1:** Comparisons between different field campaigns for particle pH, major inorganic ions
and gases and meteorological conditions. All pH values were calculated using ISORROPIA-II
run in forward mode. These statistics were previously compiled by Guo et al. (2017a). Campaign
acronyms used here stand for the California Research at the Nexus of Air Quality and Climate
Change (CalNex), Southern Oxidant and Aerosol Study (SOAS), and Southeastern Nexus of Air
Quality and Climate (SENEX).

| Campaign | CalNex | | SOAS | SENEX | This study |
|---|---|---|---|---|---|
| Type | Ground | | Ground | Aircraft | Ground |
| PM cut size | $PM_1$ | $PM_{2.5}$[a] | $PM_1$&$PM_{2.5}$[b] | $PM_1$ | $PM_1$ |
| Year | 2010 | | 2013 | 2013 | 2016 |
| Season | (Early Summer) | | Summer | Summer | Fall |
| Region/Location | SW US | | SE US | SE US | SE US |
| $SO_4^{2-}$, µg m$^{-3}$ | 2.86 ± 1.70 | 1.88 ± 0.69 | 1.73 ± 1.21 | 2.05 ± 0.80 | 1.6 ± 0.4 |
| $NO_3^-$, µg m$^{-3}$ | 3.58 ± 3.65 | 3.74 ± 1.53 | 0.08 ± 0.08 | 0.28 ± 0.09 | 0.20 ± 0.10 |
| $HNO_3$, µg m$^{-3}$ | 6.65 ± 7.03 | 4.45 ± 3.59 | 0.36 ± 0.14 | 1.35 ± 0.66 | 0.50 ± 0.26 |
| $\varepsilon(NO_3^-)$ | 39 ± 16 % | 51 ± 18 % | 22 ± 16 % | 18 ± 6 % | 26 ± 15 % |
| Total $NO_3^-$, µg m$^{-3}$ | 10.22 ± 9.74 | 8.19 ± 3.89 | 0.45 ± 0.26 | 1.63 ± 0.70 | 0.70 ± 0.28 |
| $NH_4^+$, µg m$^{-3}$ | 2.06 ± 1.67 | 1.79 ± 0.65 | 0.46 ± 0.34 | 1.06 ± 0.25 | 0.40 ± 0.20 |
| $NH_3$, µg m$^{-3}$ | 1.37 ± 0.90 | 0.75 ± 0.61 | 0.39 ± 0.25 | 0.12 ± 0.19 | 5.79 ± 3.67 |
| $\varepsilon(NH_4^+)$ | 55 ± 25% | 71 ± 19% | 50 ± 25% | 92 ± 11% | 7 ± 5 % |
| Total $NH_4^+$, µg m$^{-3}$ | 3.44 ± 1.81 | 2.54 ± 0.89 | 0.78 ± 0.50 | 1.17 ± 0.81 | 6.19 ± 3.68 |
| $Na^+$, µg m$^{-3}$ | \ | 0.77 ± 0.39 | 0.03 ± 0.07 | \ | \ |
| $Cl^-$, µgm$^{-3}$ | \ | 0.64 ± 0.48 | 0.02 ± 0.03 | \ | 0.01 ± 0.01 |
| RH, % | 79 ± 17 | 87 ± 9 | 74 ± 16 | 72 ± 9 | 69 ± 18 |
| T, °C | 18 ± 4 | 18 ± 3 | 25 ± 3 | 22 ± 3 | 24 ± 4 |
| $W_i$, µg m$^{-3}$ | 13.9 ± 18.1 | 29.8 ± 20.7 | 5.1 ± 3.8 | 3.2 ± 2.8 | 1.6 ± 1.7 |
| pH | 1.9 ± 0.5 | 2.7 ± 0.3 | 0.9 ± 0.6 | 1.1 ± 0.4 | 2.2 ± 0.6 |
| Reference | (Guo et al., 2017a) | | (Guo et al., 2015) | (Xu et al., 2016) | This study |

[a]Only during the last week of CalNex.
[b]$PM_{2.5}$ was sampled in the first half and $PM_1$ sampled in the second half of the study. Various
parameters were similar in both cases. Crustal components were higher, but are overall generally
in low concentrations so the differences had minor effects. For example, $PM_{2.5}$ $Na^+$ was 0.06 ±
0.09 µg m$^{-3}$ and $PM_1$ $Na^+$ was 0.01 ± 0.01 µg m$^{-3}$.