# Peer review of "Characterization of Aerosol Composition, Aerosol Acidity and Organic Acid Partitioning at an Agriculture-Intensive Rural Southeastern U.S. Site"

_Atmospheric Chemistry and Physics, 2018_

## Referee Comment (RC1) · Anonymous Referee #1 · 21 May 2018

Nah et al. presented a detailed case study on aerosol composition, acidity, and organic acid partitioning at an agricultural rural site in the USA. The most important contribution to the current literature is the measurement and analysis of different organic acids. The manuscript is generally well written, however, the following comments should be well addressed before consideration of publication.

These observational data sets should be very useful for the researchers who are interested in the topic of aerosol acidity and organic acid partitioning (especially there are several observations still without reasonable explanations, such as the partitioning of formic and acetic acids). Thus, if possible, I suggest these valuable obser-

vational data can be made available/accessible to the research community. In fact, the journal says that "Authors are required to provide a statement on how their underlying research data can be accessed. This must be placed as the section "Data availability" at the end of the manuscript before the acknowledgements. Please see the manuscript composition for the correct sequence. If the data are not publicly accessible, a detailed explanation of why this is the case is required. The best way to provide access to data is by depositing them (as well as related metadata) in reliable public data repositories, assigning digital object identifiers, and properly citing data sets as individual contributions." https://www.atmospheric-chemistry-and-physics.net/about/data_policy.html#data_availability

Section 2.2, Lines 262-263: the unit of Haq+ should be mole kg-1 and, given the Equation 1, the pH definition is based on molality rather than molarity.

Section 3.3, Lines 424-425: it is mentioned that "diurnal variation in particle pH is driven by Wi". Can the authors provide a quantitative analysis to show the relationship in the pH and Wi diurnal variations? I feel the Wi may not be the dominant factor that affects the diurnal variation of pH.

Section 3.3, Lines 429-439: the average pH of this study is 2.2, which is 0.3 pH unit higher than PM1 pH in CalNex. The NH3 level in this study is four times compared to CalNex. A 0.6 pH unit difference is expected from the relationship of 1 pH unit increase $\sim$ 10 times increase in NH3. This manuscript attributes this 0.3 unit difference to much higher levels of sulfate and nitrate in CalNex. I think this statement is not well justified since the ambient temperature and RH in these two campaigns are also different. I suggest to provide a more thorough analysis on this pH difference or remove these sentences from the manuscript.

Section 3.4: On the diurnal variations of organic acids. Several factors (such as emission sources and photochemical production) are provided to explain the diurnal variations of the gas-phase and particle-phase organic acids. The authors seem to ignore

the role of phase partitioning on the diurnal variations of the organic acids. If the organic acids are in a gas-particle equilibrium, no matter how they are formed, they would be re-partitioned between these two phases depending on the pH value and the aerosol water mass.

Section 3.5, Lines 605-608: an increase from 81% to 89% is expected from the S curve analysis, and what are the corresponding values ($\varepsilon C2O4(2-)$) for the observations? Do the observations support the S curve analysis?

Section 3.5, Lines 618-620: it reads from Figure S12 that there is a negative bias of $\varepsilon C2O4(2-)$ during the daytime and a positive bias during the nighttime. Can the authors provide a more quantitative analysis for the diurnal variations of $\varepsilon C2O4(2-)$?

Section 3.5, Lines 625-627: I do not think the statement that "S curves can be used to estimate activity coefficients based on gas-particle partitioning data xxx" can be derived from the data analysis in this section. For example, in Equation 4, the relationship between $\varepsilon C2O4(2-)$ and pH depends on three activity coefficients: those of H+, C2H2O4, and C2HO4-, and this relationship is nonlinear. In this case, it seems unlikely to obtain a reasonable value for any activity coefficient.
* * *

---

## Referee Comment (RC2) · Anonymous Referee #2 · 18 Jun 2018

Reference: acp_2018_373

Characterization of aerosol composition, aerosol acidity and organic acid partitioning at an agriculture-intensive rural southeastern U.S. site

The authors presented aerosol and gas measurements in an agricultural-intensive region in the southeastern U.S. during the fall season of 2016. This paper demonstrated that how particle acidity was affected by NH3 and secondary organic aerosol formation. The manuscript reveals the findings of the influence of NH3 on particle acidity and secondary organic aerosol formation through the gas to particle partitioning of semi-volatile organic acids. The topic of this manuscript is well within the scope of ACP. The

manuscript is scientifically sound and should be accepted for publication after revision. The comments are in the following.

Line 17-40: I feel that the authors described mostly the summary of the results in the abstract. I strongly recommend highlighting important findings of the study in this section.

Line 22: Define "SOA".

Line 26 and 27: What do authors mean by "study average". I think it is enough to write only "average" in the entire manuscript.

Line 32 and 33: I suggest to move the sentence "particle-phase. . . . . . . . . . .molar concentration" in the methods or results and discussion section.

Line 42-47: There are specific salts produced by the reaction of ammonia with sulfuric acid and nitric acid based on the meteorological conditions. I suggest the authors to briefly explain these points in the introduction.

Line 53-55: Although the references have been provided to back up the sentence, I suggest to at least briefly describe how ammonia is produced by industrial and vehicular emission.

Line 322 and 323: Authors found that ammonia concentration decreased at 14:00 about 1 hour before temperature decreased. Do you measure the boundary layer height during the campaign as mentioned that the decrease in ammonia concentration was because of the change in the boundary layer height?

Line 368-370: This study found that PILS and filter-based measurements of sulfate is two times lower than that of HR-TOF-AMS measurement. The similar results are also observed for nitrate and ammonium. I suggest clarifying the reason behind this in the revised manuscript.

Line 386-398: This is an exciting result that the organic aerosol mass concentration

was higher at nighttime. This result probably indicated the unique atmospheric processing of organic aerosols in the nighttime. Nevertheless, I did not find any discussion on the formation mechanisms of organic aerosols at nighttime. I suggest discussing this point in the revised version.

Line 395-399: Authors found that nitrate concentration was increased after sunset and peaked at sunrise due to the formation of organic nitrates from the nighttime chemistry of nitrate and increased gas to particle partitioning of organic and inorganic nitrates due to the decrease in temperate. I do not agree to include the sentence that the result of organic nitrates will be discussed in a future publication. The conclusion stated in lines 396 and 397 does not make sense without the data of organic nitrates.

Line 405: Which temperature and relative humidity data were used for ISORROPIA-II model inputs? Is it fixed temperature and RH values or temperature and RH observed during the campaign?

Line 433 and 435: What do authors mean by highly hygroscopic? Are that sulfate and nitrate alone contribute more to the hygroscopicity of aerosol particles or their salts play a role in the hygroscopic behavior of aerosol particles?

Line 518-520: Which type of biogenic volatile organic compound precursors elevated at high temperature and produced a high amount of organic acids during warm and sunny days? Can you give some examples?

Line 539-542: Glutarate is a higher homologous diacid of oxalate that has almost similar sources and formation processes. What is the reason that oxalate as well as malonate and succinate peaked in the mid to late afternoon but glutarate peaked on the mid-morning? I do not agree with the authors explanation that they have different biogenic volatile organic compound precursors or different production mechanisms. What about the photodegradation of glutarate to lower carbon number diacids during the afternoon?

---

## Author Comment (AC1) · 17 Jul 2018

We greatly value the careful reading and the detailed comments provided by the referees. The responses to the comments of the referees in our direct reply (shown below) and within the revised manuscript (see marked copy) are provided. The pages and lines indicated below correspond to those in the marked copy.

Please also note the supplement to this comment:
https://www.atmos-chem-phys-discuss.net/acp-2018-373/acp-2018-373-AC1-supplement.pdf

[Figure]

**Supplement:**

We greatly value the careful reading and the detailed comments provided by the referees. The responses to the comments of the referees in our direct reply (shown below) and within the revised manuscript (see marked copy) are provided. The pages and lines indicated below correspond to those in the marked copy.

**Response to Referee 1 (Referees' comments are italicized)**

1. Referee comment: "*These observational data sets should be very useful for the researchers who are interested in the topic of aerosol acidity and organic acid partitioning (especially there are several observations still without reasonable explanations, such as the partitioning of formic and acetic acids). Thus, if possible, I suggest these valuable observational data can be made available/accessible to the research community. In fact, the journal says that "Authors are required to provide a statement on how their underlying research data can be accessed. This must be placed as the section "Data availability" at the end of the manuscript before the acknowledgements. Please see the manuscript composition for the correct sequence. If the data are not publicly accessible, a detailed explanation of why this is the case is required. The best way to provide access to data is by depositing them (as well as related metadata) in reliable public data repositories, assigning digital object identifiers, and properly citing data sets as individual contributions.*"

**Author response:** The data is made available upon request:

**Page 27 line 841: "Data can be accessed by request (rweber@eas.gatech.edu)."**

2. Referee comment: "*Section 2.2, Lines 262-263: the unit of Haq+ should be mole kg-1 and, given the Equation 1, the pH definition is based on molality rather than molarity.*"

**Author response:** The referee is correct in stating that the pH definition is based on molality rather than molarity, as recommended by IUPAC. However, most thermodynamic equilibrium models (e.g., ISORROPIA-II, E-AIM) report species in terms of concentration per volume of air (e.g., μg m$^{-3}$, μmol m$^{-3}$). In addition, the particle pH can be calculated in terms of molarity, using the concentrations of species expressed in terms of molarity (mol L$^{-1}$) and concentrations per volume of air (e.g., μg m$^{-3}$) as shown in previous studies. To remove any confusion, the following changes have been made to the manuscript:

**Page 9 line 276: "The pH of an aqueous solution is defined as the negative logarithm of the hydronium ion (H$_3$O$^+$) activity on a molality basis (www.goldbook. iupac.org/html/P/P04524.html, last access: 6 July 2018):**

$$pH = -\log_{10}[a(H^+)] = -\log_{10}[m(H^+)\gamma_m(H^+)/m^\theta] \qquad \text{(1a)}$$

**where $a(H^+)$ is the hydronium ion activity in an aqueous solution, $m(H^+)$ is the hydronium ion molality, $\gamma_m(H^+)$ is the molality-based hydronium ion activity coefficient, and $m^\theta$ is the standard molality (1 mol kg$^{-1}$). For simplicity, H$_3$O$^+$ is denoted here as H$^+$ even though we recognize that the unhydrated hydrogen ion is rare in aqueous solutions. Since most thermodynamic equilibrium models (e.g., ISORROPIA-II, E-AIM) report species in terms of concentration per volume of air (e.g., μg m$^{-3}$, μmol m$^{-3}$), the particle pH can be calculated as:**

$$pH = -\log_{10} \gamma_{H^+} H_{aq}^+ = -\log_{10} \frac{1000 \gamma_{H^+} H_{air}^+}{W_i + W_o} \cong -\log_{10} \frac{1000 \gamma_{H^+} H_{air}^+}{W_i} \quad \text{(1b)}$$

where $\gamma_{H^+}$ is the molarity-based hydronium ion activity coefficient (assumed to be 1), $H_{aq}^+$ (mole L$^{-1}$) is the molar concentration of hydronium ions in particle water (i.e., pH is calculated in terms of molarity), $H_{air}^+$ (µg m$^{-3}$) is the hydronium ion concentration per volume of air, and $W_i$ and $W_o$ (µg m$^{-3}$) are the bulk particle water concentrations associated with inorganic and organic species per volume of air, respectively. In equation 1b, the molecular weight of H$^+$ is taken as 1 g mole$^{-1}$, and 1000 is the factor needed for unit conversion of g L$^{-1}$ to µg m$^{-3}$."

3. Referee comment: *"Section 3.3, Lines 424-425: it is mentioned that "diurnal variation in particle pH is driven by Wi". Can the authors provide a quantitative analysis to show the relationship in the pH and Wi diurnal variations? I feel the Wi may not be the dominant factor that affects the diurnal variation of pH."*

**Author response:** The referee is correct in stating that $W_i$ is not the dominant factor that affects the diurnal variation of particle pH. Further analysis of the diurnal profiles of $W_i$ and $H_{air}^+$ reveals that their maximum/minimum ratios are comparable (6.5 and 5.3, respectively). This indicates that the diurnal variation of particle is driven by both $W_i$ and $H_{air}^+$. This information has been added to the revised manuscript:

**Page 15 line 475: "PM$_1$ pH varied by approximately 1.4 units throughout the day. $W_i$ has an average value of 1.6 ± 1.7 µg m$^{-3}$. PM$_1$ $W_i$ and pH showed similar diurnal profiles, with both peaking in the mid-morning and reaching their minima in the mid-afternoon. These diurnal trends are consistent with those previously reported by Guo et al. (2015) for PM$_1$ measured during the summer and winter in different parts of the southeastern U.S. Also shown in Fig. 3b is the diurnal profile of $H_{air}^+$, which peaked in the mid-afternoon. The $W_i$ and $H_{air}^+$ maximum/minimum ratios are comparable (6.5 and 5.3, respectively), thus indicating that the diurnal variation in particle pH is driven by both $W_i$ and $H_{air}^+$."**

[Figure]

**Figure 3: (a) Time series and (b) diurnal profiles of ISORROPIA-predicted PM$_1$ pH and $W_i$. The diurnal profiles of RH and ISORROPIA-predicted $H_{air}^+$ are also shown in panel (b). Dates and times displayed are local time. All the data shown here represent averages in 1-hour intervals. Error bars shown in panel (b) are the standard errors.**

4. Referee comment: "*Section 3.3, Lines 429-439: the average pH of this study is 2.2, which is 0.3 pH unit higher than PM1 pH in CalNex. The NH3 level in this study is four times compared to CalNex. A 0.6 pH unit difference is expected from the relationship of 1 pH unit increase ~ 10 times increase in NH3. This manuscript attributes this 0.3 unit difference to much higher levels of sulfate and nitrate in CalNex. I think this statement is not well justified since the ambient temperature and RH in these two campaigns are also different. I suggest to provide a more thorough analysis on this pH difference or remove these sentences from the manuscript.*"

**Author response:** The referee is correct in stating that meteorological differences, specifically ambient RH and temperature, in this study vs. the CalNex campaign may also contribute to the 0.3 pH unit difference. As such, we have revised the manuscript to be more circumspect about the role of PM$_1$ NO$_3^-$ mass concentrations in the CalNex campaign causing the 0.3 pH unit difference:

**Page 16 line 498: "This may be due, in part, to PM$_1$ SO$_4^{2-}$ and NO$_3^-$ mass concentrations at CalNex being approximately 2 times and 18 times larger than those of this study, respectively. Aerosol inorganic SO$_4^{2-}$ and NO$_3^-$ species are hygroscopic species. The much higher NO$_3^-$ mass concentrations in the CalNex campaign (due, in part, to high NO$_x$ emissions) increased particle $W_i$ substantially, which diluted H$^+$ and raised particle pH, resulting in more gas-to-particle partitioning of NO$_3^-$, and eventually leading to pH levels similar to those observed in this study. This type of feedback does not happen in the southeastern U.S. where non-volatile SO$_4^{2-}$ dominates the uptake of particle water. It is also possible that the higher RH and lower**

**temperatures during the CalNex campaign (relative to this study) contributed to high particle $W_i$, which diluted H$^+$ and raised particle pH levels similar to those observed in this study."**

5. Referee comment: "*Section 3.4: On the diurnal variations of organic acids. Several factors (such as emission sources and photochemical production) are provided to explain the diurnal variations of the gas-phase and particle-phase organic acids. The authors seem to ignore the role of phase partitioning on the diurnal variations of the organic acids. If the organic acids are in a gas-particle equilibrium, no matter how they are formed, they would be re-partitioned between these two phases depending on the pH value and the aerosol water mass.*"

**Author response:** The referee is correct in stating that gas-particle partitioning of organic acids can also contribute to the observed diurnal variations of the organic acids. As such, this possibility has been added to the revised manuscript:

**Page 19 line 603: "Some of these gas-phase organic acids may also be formed in the particle phase during organic aerosol photochemical aging, with subsequent volatilization into the gas phase. The gas-particle partitioning of organic acids likely depends on thermodynamic conditions, which are controlled by particle pH and $W_i$ and meteorological conditions as will be shown in section 3.5."**

6. Referee comment: "*Section 3.5, Lines 605-608: an increase from 81% to 89% is expected from the S curve analysis, and what are the corresponding values ($\varepsilon C2O4(2-)$) for the observations? Do the observations support the S curve analysis?*"

**Author response:** Although the measured $\varepsilon(C_2O_4^{2-})$ values are generally consistent with the calculated S curve, the measured $\varepsilon(C_2O_4^{2-})$ data at particle pH 2.2 and 2.5 is not clear enough to give definite values to support S curve analysis due to data scatter. For example, the averages ($\pm$ standard deviation) of the measured $\varepsilon(C_2O_4^{2-})$ at pH 2.2 and pH 2.5 are similar (78 $\pm$ 3 % and 79 $\pm$ 3 %, respectively) because of the scatter. Therefore, we emphasize in the revised manuscript that the S curve can be used to gain a qualitative understanding of how high NH$_3$ events at the site affect oxalic acid gas-particle partitioning:

**Page 22 line 704: "Since the measured $\varepsilon(C_2O_4^{2-})$ are in general agreement with the analytically calculated S curve (Fig. 7), we can use the S curve to understand qualitatively how high NH$_3$ events at the site affect oxalic acid gas-particle partitioning. Here we define high NH$_3$ events as periods where the NH$_3$ concentration was higher than 13.3 ppb (which is the average NH$_3$ concentration + 1 standard deviation). As discussed in section 3.3, the PM$_1$ pH during high NH$_3$ events is 2.5 $\pm$ 0.6, which is slightly higher than the average PM$_1$ pH of 2.2 $\pm$ 0.6. Based on the S curve calculated using the average temperature and $W_i$ values, $\varepsilon(C_2O_4^{2-})$ increases from 81 % to 89 % when particle pH increases from 2.2 to 2.5. While this result indicates that high NH$_3$ concentrations can raise the particle pH sufficiently such that it can promote gas-to-particle partitioning of oxalic acid, this is not always the case. Specifically, increasing the particle pH from -2 (or lower) to 1 will not result in a significant increase in $\varepsilon(C_2O_4^{2-})$. Therefore, whether or not particle pH, and consequently oxalic acid gas-particle partitioning, is sensitive to NH$_3$ concentration depends strongly on particle pH values."**

7. Referee comment: "*Section 3.5, Lines 618-620: it reads from Figure S12 that there is a negative bias of εC2O4(2-) during the daytime and a positive bias during the nighttime. Can the authors provide a more quantitative analysis for the diurnal variations of εC2O4(2-)?*"

**Author response:** Note that Fig. S12 in the original manuscript is now Fig. S13 in the revised manuscript. First, we view this change in partitioning resulting from changes in temperature and particle $W_i$ as a shift, not a bias. Second, the purpose of Fig. S13 is to show that the S curve will change during the transition from day to night as a result of changes in meteorological conditions and particle $W_i$. Ambient RH and temperatures are higher and lower at night, respectively. Particle $W_i$ will increase as a result of these changes in RH and temperature during the transition from day to night. These changes in the meteorological conditions and particle $W_i$ will generally result in a higher fraction of oxalic acid partitioning to the particle phase for particle pH in this study. Since this shift is non-linear (i.e., see changes in S curve shape), we feel the best way to show the changes is graphically, as done in Fig. S13. It is not possible to provide a more quantitative analysis of $\varepsilon(C_2O_4{}^{2-})$.

8. Referee comment: "*Section 3.5, Lines 625-627: I do not think the statement that "S curves can be used to estimate activity coefficients based on gas-particle partitioning data xxx" can be derived from the data analysis in this section. For example, in Equation 4, the relation- ship between εC2O4(2-) and pH depends on three activity coefficients: those of H+, C2H2O4, and C2HO4-, and this relationship is nonlinear. In this case, it seems unlikely to obtain a reasonable value for any activity coefficient.*"

**Author response:** We agree with the referee's point. Therefore, we have removed the above-mentioned statement in the revised manuscript.

**Response to Referee 2 (Referees' comments are italicized)**

1. Referee comment: "*Line 17-40: I feel that the authors described mostly the summary of the results in the abstract. I strongly recommend highlighting important findings of the study in this section.*"

**Author response:** We respectfully disagree with the referee's comment that the abstract is mostly a summary of the results. There are five important findings of this study, and they have been highlighted in the abstract:

1) Despite the high $NH_3$ concentrations (average $8.1 \pm 5.2$ ppb), $PM_1$ were highly acidic with pH values ranging from 0.9 to 3.8, and an average pH of $2.2 \pm 0.6$.

2) The measured molar fraction of oxalic acid in the particle phase (i.e., particle-phase oxalic acid molar concentration divided by the total oxalic acid molar concentration) ranged between 47 and 90 % for $PM_1$ pH 1.2 to 3.4.

3) The measured oxalic acid gas-particle partitioning ratios were in good agreement with their corresponding thermodynamic predictions, calculated based on oxalic acid's physicochemical properties, ambient temperature, particle water and pH.

4) The measured formic and acetic acid gas-particle partitioning ratios did not agree with their corresponding thermodynamic predictions.

5) Our study suggests that while higher $NH_3$ concentrations may lead to higher organic aerosol mass concentrations due to increased gas-to-particle partitioning of some organic acids, this effect is minor since organic acids comprised a small fraction of the overall aerosol mass.

2. Referee comment: "*Line 22: Define "SOA".*"

**Author response:** This is defined in the revised manuscript.

3. Referee comment: "*Line 26 and 27: What do authors mean by "study average". I think it is enough to write only "average" in the entire manuscript.*"

**Author response:** We have replaced "study average" with "average" in the revised manuscript.

4. Referee comment: "*Line 32 and 33: I suggest to move the sentence "particle-phase. . . . . . . . . . . . . . . ..molar concentration" in the methods or results and discussion section.*"

**Author response:** The co-editor previously requested that the above-mentioned sentence be added to the abstract to prevent any confusion. Therefore, we will keep the sentence in.

5. Referee comment: "*Line 42-47: There are specific salts produced by the reaction of ammonia with sulfuric acid and nitric acid based on the meteorological conditions. I suggest the authors to briefly explain these points in the introduction.*"

**Author response:** As requested, we have added a brief explanation on how the formation of specific salts in the particle phase is dependent on environmental conditions in the revised manuscript:

**Page 2 line 48: "The formation of particle-phase ammonium sulfate and nitrate salts in the aerosol phase depends on the thermodynamic states of their precursors and the environmental conditions, which can consequently affect aerosol pH. For example, Guo et al. (2017b) showed that for Southeast U.S. summertime conditions, as aerosol pH increases, the relative fractions of $SO_4^{2-}$ and $HSO_4^-$ increases and decreases, respectively."**

**Author response:** Disagreements between the HR-ToF-AMS and PILS and filter-based measurements are due to our application of composition-dependent collection efficiency (CDCE) values to the raw HR-ToF-AMS data. In our previous manuscript, we calculated CDCE values using the CDCE parameterization method proposed by Middlebrook et al. (2012), which derives CE values based largely on aerosol inorganic species concentrations and the relative humidity in the sampling line. Under our sampling conditions, the Middlebrook parameterization method estimated CDCE values of 0.44 to 0.55 (average of 0.45). However, the application of these CDCE values to the raw HR-ToF-AMS data resulted in the $SO_4^{2-}$, $NO_3^-$ and $NH_4^+$ measurements being higher than the PILS and filter-based measurements. This is likely due to organics dominating the aerosol composition during the study (average of 74.2 ± 7.9 % of the non-refractory $PM_1$ mass concentration). Lee et al. (2015) suggested that a high organic mass fraction may hinder the complete efflorescence of aerosols when they are passed through the drier prior to delivery into the HR-ToF-AMS, reducing the particle bounce and increasing the CE value. As described in our previous manuscript, the CDCE-corrected HR-ToF-AMS measurements had to be scaled by a constant factor of 0.5 in order for them to agree with the PILS and filter-based measurements. It should be noted a previous ambient study also reported poor agreement between the CDCE-corrected HR-ToF-AMS measurements and parallel aerosol composition measurements due to high organic aerosol mass concentrations (see Lee et al. (2015)).

For these reasons, we applied a constant CE value of 0.9 to the raw HR-ToF-AMS data. This CE value was determined from comparisons of the raw HR-ToF-AMS data with PILS measurements. To remove any confusion, the following changes have been made to the manuscript:

**Page 6 line 187: "Composition-dependent collection efficiency (CDCE) values of 0.44 to 0.55 were determined using the procedure detailed by Middlebrook et al. (2012), where CDCE values are derived based largely on aerosol inorganic species concentrations and the relative humidity in the sampling line. In addition, a constant collection efficiency (CE) value of 0.9 was determined from the comparison of raw HR-ToF-AMS $SO_4^{2-}$ data with other particulate $SO_4^{2-}$ measurements performed during the study. Comparisons of aerosol mass concentrations obtained from the application of CDCE values (i.e., 0.44 to 0.55) vs. a constant CE value (i.e., 0.9) to the raw HR-ToF-AMS data are discussed in section 3.2."**

**Page 13 line 389: "The aerosol inorganic chemical composition was measured by several instruments during this study. The HR-ToF-AMS, PILS-IC and PILS-HPIC measured the composition of $PM_1$, while a filter-based particle composition monitor measured the composition of $PM_{2.5}$. Comparisons of aerosol $SO_4^{2-}$, $NO_3^-$ and $NH_4^+$ mass concentrations obtained from the application of CDCE values to the raw HR-ToF-AMS data are compared to those measured by the other three instruments in Fig. S6. $NH_4^+$ measurements by the PILS-IC are not available for comparison due to denuder breakthrough that occurred**

during the study.

$SO_4^{2-}$ measurements by the various instruments are generally well correlated with each other, with $R^2$ values ranging from 0.64 to 0.92. Although $PM_1$ $SO_4^{2-}$ measurements by the two PILS systems show good agreement with each other, HR-ToF-AMS CDCE-applied $SO_4^{2-}$ measurements are approximately two times higher than the PILS and filter measurements. Similar systematic differences are also observed for $NO_3^-$ and $NH_4^+$ measurements. $NO_3^-$ and $NH_4^+$ measurements by the four instruments are moderately correlated ($R^2 = 0.54$ to 0.79 and $R^2 = 0.94$, respectively). $NO_3^-$ measurements by the PILS and filter systems are mostly similar; however, HR-ToF-AMS CDCE-applied $PM_1$ $NO_3^-$ and $NH_4^+$ measurements are approximately three times and two times higher than the PILS and filter measurements. One possible reason is that the calculated CDCE is lower due to organics dominating the aerosol composition during the study (average of 74.2 ± 7.9 % of the non-refractory $PM_1$ mass concentration). Lee et al. (2015) suggested that a high organic mass fraction may impede the complete efflorescence of aerosols when they are passed through the drier prior to delivery into the HR-ToF-AMS, thus reducing the particle bounce and increasing the CE value. Hence, we estimated HR-ToF-AMS $PM_1$ mass concentrations that would be consistent with PILS and filter measurements by multiplying all the raw HR-ToF-AMS data by a constant CE value of 0.9, which was obtained from comparisons of the raw HR-ToF-AMS $SO_4^{2-}$ data with PILS-IC and PILS-HPIC $SO_4^{2-}$ measurements. The constant CE-applied HR-ToF-AMS data is used in all our subsequent analyses.

Figure S6 caption: "Aerosol (panels a to d) $SO_4^{2-}$, (panels e to h) $NO_3^-$, and (i) $NH_4^+$ comparisons between HR-ToF-AMS, PILS-IC, PILS-HPIC and filters for the entire field study. CDCE values were applied to the raw HR-ToF-AMS data to obtain the mass concentrations shown here (see main text for details). For comparisons between the HR-ToF-AMS, PILS-IC and PILS-HPIC data (panels c, d, g and h), the measurements are averaged over 1 hour intervals. For comparisons with filter data (panels a, b, e, f and i), the HR-ToF-AMS, PILS-IC and PILS-HPIC data are averaged over 24 hour intervals. Orthogonal regression fits are shown. Uncertainties in the fits are 1 standard deviation."

[revised manuscript text omitted]
. CDCE values were applied to the raw HR-ToF-AMS data to obtain the mass concentrations shown here (see main text for details). For comparisons between the HR-ToF-AMS, PILS-IC and PILS-HPIC data (panels c, d, g and h), the measurements are averaged over 1 hour intervals. For comparisons with filter data (panels a, b, e, f and i), the HR-ToF-AMS, PILS-IC and PILS-HPIC data are averaged over 24

hour intervals. Orthogonal regression fits are shown. Uncertainties in the fits are 1 standard deviation.

[Figure]

**Figure S7:** Diurnal profiles of the total nitrate functionality contributed by organic and inorganic nitrates ($NO_{3,meas}$), and the nitrate functionality solely from organic nitrates ($NO_{3,org}$) and inorganic nitrates ($NO_{3,inorg}$). $NO_{3,org}$ and $NO_{3,inorg}$ are estimated using the $NO^+/NO_2^+$ ratio method as described by Farmer et al. (2010) and Xu et al. (2015). Similar to Xu et al. (2015), we used a $R_{ON}$

(defined here as the $NO^+/NO_2^+$ ratio for organic nitrates) value of 10 to calculate $NO_{3,org}$ and

[revised manuscript text omitted]

---

## Author Response (AR3)

We greatly value the careful reading and the detailed comments provided by the referees. The responses to the comments of referee 1 in our direct reply (shown below) and within the revised manuscript (see marked copy) are provided. The pages and lines indicated below correspond to those in the marked copy.

**Response to Referee 1 (Referees' comments are italicized)**

1. Referee comment: "*The authors have well addressed my comments for an earlier version of manuscript except for the one on pH definition. The revised manuscript says that "rH+ is the molarity-based hydronium ion activity coefficient (assumed to be 1), Haq+ (mole L-1) is the molar concentration of hydronium ions in particle water (i.e., pH is calculated in terms of molarity)". My understanding is that the molarity (or molar concentration) is defined on the basis of volume of the solution instead of that of the water. Haq+ calculated by ISORROPIA is actually the molality (mole kg-1 water). Please correct me if I was wrong. See more here: https://en.wikipedia.org/wiki/Molar_concentration*"

**Author response:** The referee is correct, and we have changed the manuscript to state that it is molality-based:

**Page 10 line 273: "Since most thermodynamic equilibrium models (e.g., ISORROPIA-II, E-AIM) do not report liquid concentrations, but instead report species in terms of concentration per volume of air (e.g., µg m$^{-3}$, µmol m$^{-3}$), we have calculated the particle pH by:**

$$pH = -\log_{10} \gamma_{H^+} H_{aq}^+ = -\log_{10} \frac{1000 \gamma_{H^+} H_{air}^+}{W_i + W_o} \cong -\log_{10} \frac{1000 \gamma_{H^+} H_{air}^+}{W_i} \quad \text{(1b)}$$

**where $\gamma_{H^+}$ is the hydronium ion activity coefficient (assumed to be 1), $H_{aq}^+$ is the concentration of hydronium ions in particle water in mole L$^{-1}$ (i.e., the density of water is assumed to be 1000 kg m$^{-3}$, and so pH is calculated in terms of molality), $H_{air}^+$ (µg m$^{-3}$) is the hydronium ion concentration per volume of air, and $W_i$ and $W_o$ (µg m$^{-3}$) are the bulk particle water concentrations associated with inorganic and organic species per volume of air, respectively. In equation 1b, the molecular weight of H$^+$ is taken as 1 g mole$^{-1}$, and 1000 is the factor needed for unit conversion of g L$^{-1}$ to µg m$^{-3}$. $H_{air}^+$ and $W_i$ are outputs of the ISORROPIA-II model."**

The referee is correct in stating that $H_{aq}^+$ has units of molality (mol kg$^{-1}$). However, $H_{aq}^+$ is not reported by ISORROPIA but is defined in equation 1b. As stated in the manuscript, ISORROPIA report species in terms of concentration per volume of air (e.g., µg m$^{-3}$, µmol m$^{-3}$). $H_{air}^+$ and $W_i$ are outputs of the ISORROPIA model, and they are expressed in terms of µg m$^{-3}$ by ISORROPIA. In using $H_{air}^+$ and $W_i$ in Equation 1b to calculate the aerosol pH, we are defining $H_{aq}^+$ as a molality-based concentration.

We refer the referee to page 8 of the User's Manual found on the ISORROPIA website (http://isorropia.eas.gatech.edu/index.php?title=User%27s_Manual) for more information regarding the units used by ISORROPIA.

2. Referee comment: *"I have another minor comment on this change made in the revised manuscript: Page 25 line 791: "In addition, formic and acetic acids may not be internally mixed with most of the other PM1 aerosol components (e.g., SO42-, NO3-, NH4+, CH3CO2H), and thus are not associated with acidic aerosols, as assumed above." What does "CH3CO2H" mean here?"*

**Author response:** $CH_3COOH$ is the chemical formula of acetic acid. We meant to write $C_2O_4^{2-}$, chemical formula of oxalate, which has been shown to be internally mixed with $PM_1$ aerosol components $SO_4^{2-}$, $NO_3^-$, $NH_4^+$ in section 3.5.1 of our manuscript. This is corrected in the revised manuscript:

**Page 25 line 717: "In addition, formic and acetic acids may not be internally mixed with most of the other PM$_1$ aerosol components (e.g., SO$_4^{2-}$, NO$_3^-$, NH$_4^+$, C$_2$O$_4^{2-}$), and thus are not associated with acidic aerosols, as assumed above."**

**Additional minor revisions**

1. The affiliation of the first author was updated.

2. We changed the range of the x-axis of graphs shown in Figs. 4, 7, 8, S12 and S13. The x-axis of the previous version of these graphs had their range starting from pH value -2. The x-axis of the revised version of these graphs have their range starting from pH value -1. We made these changes because the particle pH of ambient aerosol does not go below -1. Note that these changes do not affect our results.

[Figure]

[revised manuscript text omitted]
. CDCE values were applied to the raw HR-ToF-AMS data to obtain the mass concentrations shown here (see main text for details). For comparisons between the HR-ToF-AMS, PILS-IC and PILS-HPIC data (panels c, d, g and h), the measurements are averaged over 1 hour intervals. For comparisons with filter data (panels a, b, e, f and i), the HR-ToF-AMS, PILS-IC and PILS-HPIC data are averaged over 24 hour intervals. Orthogonal regression fits are shown. Uncertainties in the fits are 1 standard deviation.

[Figure]

**Figure S7:** Diurnal profiles of the total nitrate functionality contributed by organic and inorganic nitrates ($NO_{3,meas}$), and the nitrate functionality solely from organic nitrates ($NO_{3,org}$) and inorganic nitrates ($NO_{3,inorg}$). $NO_{3,org}$ and $NO_{3,inorg}$ are estimated using the $NO^+/NO_2^+$ ratio method as described by Farmer et al. (2010) and Xu et al. (2015). Similar to Xu et al. (2015), we used a $R_{ON}$

(defined here as the $NO^+/NO_2^+$ ratio for organic nitrates) value of 10 to calculate $NO_{3,org}$ and

[revised manuscript text omitted]

Nah, T., Ji, Y., Tanner, D. J., Guo, H., Sullivan, A. P., Ng, N. L., Weber, R. J., and Huey, L. G.:

Real-time measurements of gas-phase organic acids using SF6- chemical ionization mass spectrometry, Atmos. Meas. Tech. Discuss., 2018, 1-40, 10.5194/amt-2018-46, 2018.

Neuman, J. A., Ryerson, T. B., Huey, L. G., Jakoubek, R., Nowak, J. B., Simons, C., and

Fehsenfeld, F. C.: Calibration and evaluation of nitric acid and ammonia permeation tubes by UV

optical absorption, Environmental Science & Technology, 37, 2975-2981, 10.1021/es0264221,

2003.

Sander, R.: Compilation of Henry's law constants (version 4.0) for water as solvent, Atmos. Chem.

Phys., 15, 4399-4981, 10.5194/acp-15-4399-2015, 2015.

Saxena, P., and Hildemann, L. M.: Water-soluble organics in atmospheric particles: A critical review of the literature and application of thermodynamics to identify candidate compounds,

Journal of Atmospheric Chemistry, 24, 57-109, 10.1007/bf00053823, 1996.

Schrier, E. E., Pottle, M., and Scheraga, H. A.: The Influence of Hydrogen and Hydrophobic Bonds on the Stability of the Carboxylic Acid Dimers in Aqueous Solution, Journal of the American

Chemical Society, 86, 3444-3449, 10.1021/ja01071a009, 1964.

Xu, L., Suresh, S., Guo, H., Weber, R. J., and Ng, N. L.: Aerosol characterization over the southeastern United States using high-resolution aerosol mass spectrometry: spatial and seasonal variation of aerosol composition and sources with a focus on organic nitrates, Atmos. Chem. Phys.,

15, 7307-7336, 10.5194/acp-15-7307-2015, 2015.

Xu, L., Guo, H. Y., Weber, R. J., and Ng, N. L.: Chemical Characterization of Water-Soluble

Organic Aerosol in Contrasting Rural and Urban Environments in the Southeastern United States,

Environmental Science & Technology, 51, 78-88, 10.1021/acs.est.6b05002, 2017.

Zuend, A., Marcolli, C., Luo, B. P., and Peter, T.: A thermodynamic model of mixed organic- inorganic aerosols to predict activity coefficients, Atmos. Chem. Phys., 8, 4559-4593,

10.5194/acp-8-4559-2008, 2008.

Zuend, A., Marcolli, C., Booth, A. M., Lienhard, D. M., Soonsin, V., Krieger, U. K., Topping, D.

O., McFiggans, G., Peter, T., and Seinfeld, J. H.: New and extended parameterization of the thermodynamic model AIOMFAC: calculation of activity coefficients for organic-inorganic
mixtures containing carboxyl, hydroxyl, carbonyl, ether, ester, alkenyl, alkyl, and aromatic
functional groups, Atmos. Chem. Phys., 11, 9155-9206, 10.5194/acp-11-9155-2011, 2011.

Zuend, A., Marcolli, C., Luo, B. P., and Peter, T.: A thermodynamic model of mixed organic-
inorganic aerosols to predict activity coefficients (vol 8, pg 4559, 2008), Atmos. Chem. Phys., 12,
10075-10075, 10.5194/acp-12-10075-2012, 2012.